# Unified Optimal Transport Framework for Universal Domain Adaptation

**Wanxing Chang**[1]     **Ye Shi**[1,3*]    **Hoang Duong Tuan**[2]    **Jingya Wang**[1,3*]

[1]ShanghaiTech University     [2]University of Technology Sydney
[3]Shanghai Engineering Research Center of Intelligent Vision and Imaging

{changwx,shiye,wangjingya}@shanghaitech.edu.cn
tuan.hoang@uts.edu.au

## Abstract

Universal Domain Adaptation (UniDA) aims to transfer knowledge from a source domain to a target domain without any constraints on label sets. Since both domains may hold private classes, identifying target common samples for domain alignment is an essential issue in UniDA. Most existing methods require manually specified or hand-tuned threshold values to detect common samples thus they are hard to extend to more realistic UniDA because of the diverse ratios of common classes. Moreover, they cannot recognize different categories among target-private samples as these private samples are treated as a whole. In this paper, we propose to use Optimal Transport (OT) to handle these issues under a unified framework, namely UniOT. First, an OT-based partial alignment with adaptive filling is designed to detect common classes without any predefined threshold values for realistic UniDA. It can automatically discover the intrinsic difference between common and private classes based on the statistical information of the assignment matrix obtained from OT. Second, we propose an OT-based target representation learning that encourages both global discrimination and local consistency of samples to avoid the over-reliance on the source. Notably, UniOT is the first method with the capability to automatically discover and recognize private categories in the target domain for UniDA. Accordingly, we introduce a new metric $H^3$-score to evaluate the performance in terms of both accuracy of common samples and clustering performance of private ones. Extensive experiments clearly demonstrate the advantages of UniOT over a wide range of state-of-the-art methods in UniDA.

## 1   Introduction

Deep neural networks have boosted performance in extensive computer vision tasks but still struggle to generalize well in cross-domain tasks that source and target domain data are drawn from the different data distributions. Unsupervised Domain Adaptation (UDA) [28] aims to transfer knowledge from fully labeled source to unlabeled target domain by minimizing the domain gap between source and target. However, existing UDA methods tackle the domain gap under a strong closed-set assumption that two domains share identical label sets, limiting their applications to real-world scenarios. Recently, Partial Domain Adaptation (PDA) [4] and Open Set Domain Adaptation (OSDA) [2] are proposed to relax the closed-set assumption, allowing the existence of private classes in the source and target domain respectively. However, all the above-mentioned settings heavily rely on

---

[*]Corresponding author.

36th Conference on Neural Information Processing Systems (NeurIPS 2022).

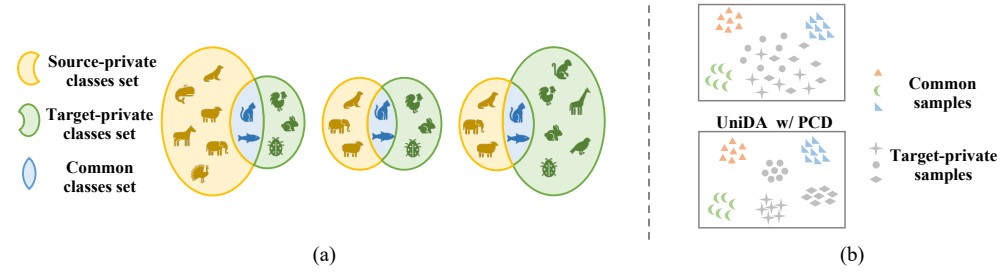

Figure 1: (a) Realistic UniDA scenarios with diverse ratios of common classes. (b) Target representation of UniDA with or without considering the Private Class Discovery (PCD).

prior knowledge where the common classes lie in the target domain. It may be not reasonable and realistic for UDA since the target domain is unsupervised.

To address the above problem, a generalized setting, termed as Universal Domain Adaptation (UniDA) [39], was proposed to allow both domains to own private classes but without knowing prior information, e.g. matched common classes and classes numbers in the target domain. Detecting common and target-private samples in the target domain is an essential issue in UniDA.

Existing UniDA methods [39, 17, 32] detect common and target-private samples by using some manually specified or hand-tuned threshold values. Therefore, these methods are not applicable to more realistic UniDA due to the diverse ratios of common categories as shown in Fig. 1(a). Moreover, most existing UniDA methods treat all target-private samples as a single class and cannot recognize different categories among them, as shown in Fig. 1(b). This paper is the first to emphasize that UniDA methods should have the capability to automatically discover and recognize private categories in the target domain.

Essentially, the common class detection and private class discovery problems can be both viewed as distribution transportation problems. Therefore, these issues can be formulated within the framework of Optimal Transport (OT), which is a promising optimization problem to seek an efficient solution for transporting one distribution to another. Even though many OT-based methods [11, 24, 38, 13] have been proposed for Unsupervised Domain Adaptation, most of them consider cross-domain sample-to-sample mapping under the closed-set condition and are not specialized in UniDA problem with unaligned label sets. Specifically, OT encourages a global mapping to mine domain statistics property for discovering intrinsic differences among common and target private samples. Additionally, the OT constraints can also avoid degenerate solutions in clustering representation problem [1, 6]. Inspired by this, OT can be a proper formulation with respect to inter-domain common class detection and intra-domain private class discovery.

In this paper, we propose a unified optimal transport framework, namely UniOT, to solve Universal Domain Adaptation from the perspectives of partial alignment for common class detection and target representation learning for private class discovery. With regards to common class detection, we propose an OT-based partial alignment to detect common samples in the target domain and develop an adaptive filling method to handle the diverse ratios of common categories. Without any predefined threshold values, it can automatically discover the intrinsic difference between common and private classes based on the global statistical information of the assignment matrix obtained from OT. For private class discovery, we propose an OT-based target representation learning that encourages both global discrimination and local consistency of samples to avoid the over-reliance on source supervision. In addition, UniOT has the capability to automatically discover and recognize categories in the target domain, benefiting from the representation learning with OT. We believe a competitive UniDA method should achieve high classification accuracy for the common class and also learn a discriminative representation for the target private class. However, existing methods do not provide any quantitative metric to evaluate the target representation performance for those unknown samples. For target-private class discovery purpose, we introduce a new evaluation metric, $H^3$-score, considering not only common class accuracy but also the clustering performance of private classes.

Our contributions are summarized as follows: (1) We propose UniOT to handle two essential issues in UniDA, including common class detection and private class discovery. To our best knowledge, this is the first attempt to jointly consider common class detection and private class discovery in a unified framework by optimal transport. (2) We propose an OT-based partial alignment with adaptive filling for common class detection without any predefined threshold values. It can automatically adapt to more realistic UniDA scenarios, where the ratios of common classes are diverse as shown in Fig. 1(a). (3) We design an OT-based representation learning technique for private class discovery, considering both global discrimination of clusters and local consistency of samples. Unlike most existing methods that treat all target-private samples as a whole, our UniOT can automatically discover and recognize private categories in the target domain.

## 2 Related Work

**Universal Domain Adaptation** As a more generalized UDA setting, UniDA [39] is more challenging and realistic since the prior information of categories is unknown. UAN [39],CMU [17] and TNT [8] designed sample-level uncertainty criteria to measure domain transferability. Samples with lower uncertainty are encouraged for adversarial adaptation with higher weights. Most UniDA methods detect common samples with the sample-level criteria, which requires some manually specified and hand-tuned threshold values. Moreover, over-reliance on source supervision under category neglects discriminative representation in the target domain. DANCE [32] proposed neighborhood clustering as a self-supervised technique to learn features useful for discriminating "unknown" categories. DCC [23] enumerated cluster numbers of the target domain to obtain optimal cross-domain consensus clusters as common classes, but the consensus clusters are not robust enough due to the hard assignment of K-Means [26]. MATHS [7] detected the existence of the target private class by Hartigan's dip test in the first stage, then trained hybrid prototypes by learning from K-Means assignment where a fixed hyper-parameter for K-Means was adopted. In this paper, we use OT to handle both common sample detection and target representation learning under a unified framework. It is worth noting that our method is adaptive for various unbalanced compositions of common and private classes, which does not require any predefined threshold. In addition, our method encourages both the global discrimination and the local consistency of samples for target representation learning.

**Optimal Transport in Domain Adaptation** Optimal transport (OT) was first proposed by Kantorovich [22] to obtain an efficient solution for moving one distribution of mass to another. Interior point methods can handle OT problem with computational complexity at least $\mathcal{O}(d^3 log(d))$. Cuturi [10] first proposed to use Sinkorn's algorithm [34] to compute an approximate transport coupling with an entropic regularization. This method is lightspeed and can handle large-scale problems efficiently. Later, DeepJDOT [11] used OT to address the domain adaptation problems to pair samples in the source and target domain. Accordingly, a new classifier is trained on the transported source data representation. Since classical OT does not allow partial displacement in transport plan, unbalanced optimal transport (UOT) is proposed to relax equality constraint by replacing the hard marginal constraints of OT with soft penalties using Kullback-Leibler (KL) divergence [16], which can also effectively be solved by generalized Sinkhorn's algorithm [9]. JUMBOT [13] proposed a minibatch strategy coupled with unbalanced optimal transport, which can yield more robust behavior for handling minibatch UDA and PDA problem. Also, TS-POT [27] addressed the partial mapping problem in UDA and PDA with partial OT [14]. However, these methods consider a cross-domain sample-to-sample mapping under the closed-set condition or label-set prior, and are not suitable for the unknown category gap problem in UniDA. Besides, a sample-to-sample mapping neglects the global consensus of two domains and intrinsic differences among common and target private samples. In this paper, we propose an inter-domain partial alignment based on UOT for common class detection and intra-domain representation learning based on OT for private class discovery, where a sample-to-prototype mapping is designed to encourage a more global-aware assignment.

**Clustering for Representation Learning** Deep clustering has raised increasing attention in deep learning community due to its capability of representation learning for discriminative clusters. DeepCluster [5] proposed an end-to-end framework that iteratively obtains clustering assignment by K-Means [26] and updates feature representation by the assignment, while PICA [21] simultaneously learns both feature representation and clustering assignment. A two-step approach was proposed by SCAN [36], which firstly learns a better initial representation, then implements an end-to-end clustering learning in the next stage. These deep clustering methods assume the number of clusters is

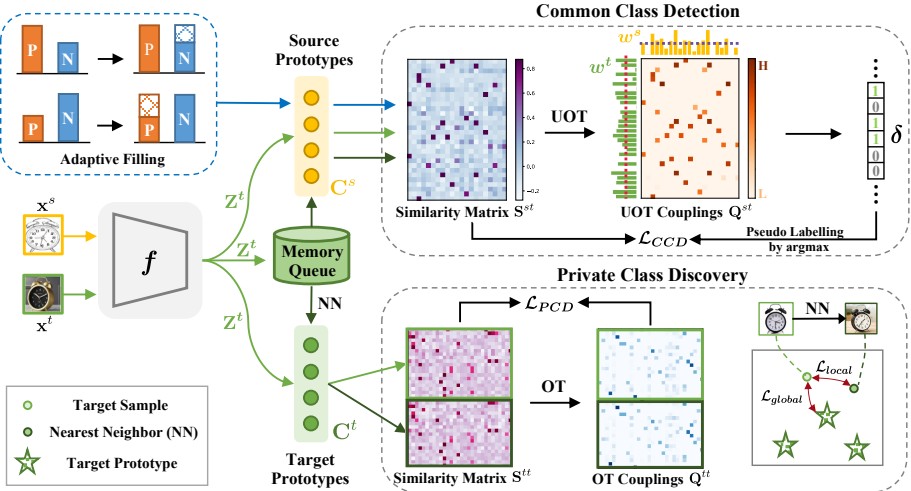

Figure 2: **An overview of the proposed method.** The UniOT mainly contains two branches: (1) Common Class Detection: We consider a partial mapping problem between target samples and source prototypes based on UOT. Highly confident target samples selected by statistics mean are pesudo-labeled as common classes for partial alignment. Adaptive Filling is developed for handling the diverse ratios of common categories. (2) Private Class Discovery: We consider a full mapping problem between target samples and target prototypes based on OT for representation learning. $\mathcal{L}_{global}$ encourages global discrimination of clusters and $\mathcal{L}_{local}$ encourages local consistency of samples.

known beforehand. Self-labelling [1] and SwAV [6] formulated the clustering assignment problem into an OT problem for better feature representation learning. Notably, the number of clusters does not rely on the ground-truth categories, benefiting from the equality constraints in OT. Therefore, OT-based representation learning is more suitable for UniDA, where the number of private class is unknown. However, existing OT-based representation learning methods [1, 6] consider global discrimination of clusters but ignore local consistency of samples, limiting direct applications in UniDA. In this paper, we develop an OT-based target representation learning technique for both better global and local structures in UniDA.

## 3 Methodology

In Universal Domain Adaptation (UniDA), there is a labeled source domain $\mathcal{D}^s = \{\mathbf{x}_i^s, \mathbf{y}_i^s\}_{i=1}^{n_s}$ and an unlabeled target domain $\mathcal{D}^t = \{\mathbf{x}_i^t\}_{i=1}^{n_t}$. We use $\mathcal{C}_s$ and $\mathcal{C}_t$ to denote the label set of source and target domain respectively. Denote $\mathcal{C} = \mathcal{C}_s \cap \mathcal{C}_t$ as the common label set shared by both domains. Let $\overline{\mathcal{C}}_s = \mathcal{C}_s \backslash \mathcal{C}$ and $\overline{\mathcal{C}}_t = \mathcal{C}_t \backslash \mathcal{C}$ represent label sets of source private and target private, respectively. UniDA aims to classify target samples into $|\mathcal{C}| + 1$ classes, where target-private samples are treated as *unknown* class uniformly.

UniDA aims to transfer knowledge from source to target under domain gap and category gap (i.e., different label sets among source and target domain), which makes it challenging to align common samples and reject private samples. Moreover, over-reliance on common knowledge extracted from the source domain neglects the target intrinsic structure. Therefore, in this paper, we propose a unified optimal transport framework to address UniDA problem from two perspectives, i.e., common class detection and private class discovery, as shown in Fig. 2.

### 3.1 Preliminary

Optimal Transport is a constrained optimization problem that seeks an efficient solution of transporting one distribution of mass to another. We now briefly recall the well-known optimal transport formulation.

Let $\Sigma_r := \{\boldsymbol{x} \in \mathbb{R}_+^r | \boldsymbol{x}^\top \mathbf{1}_r = 1\}$ denote the probability simplex, where $\mathbf{1}_d$ is a $d$-dimensional vector of ones. Given two simplex vectors $\boldsymbol{\alpha} \in \Sigma_r$ and $\boldsymbol{\beta} \in \Sigma_c$, we can write the transport polytope of $\boldsymbol{\alpha}$ and $\boldsymbol{\beta}$ as follows:

$$\mathbf{U}(\boldsymbol{\alpha}, \boldsymbol{\beta}) = \left\{ \mathbf{Q} \in \mathbb{R}_+^{r \times c} | \mathbf{Q}\mathbf{1}_c = \boldsymbol{\alpha}, \mathbf{Q}^\top \mathbf{1}_r = \boldsymbol{\beta} \right\}. \tag{1}$$

The transport polytope $\mathbf{U}(\boldsymbol{\alpha}, \boldsymbol{\beta})$ can also be interpreted as a set of all possible joint probabilities of $(X, Y)$, where $X$ and $Y$ are two $d$-dimensional random variables with marginal distribution $\boldsymbol{\alpha}$ and $\boldsymbol{\beta}$, respectively. Following SwAV [6], given a similarity matrix $\mathbf{M} \in \mathbb{R}^{r \times c}$ instead of distance matrix in [10], the coupling matrix (or joint probability) $\mathbf{Q}^*$ mapping $\boldsymbol{\alpha}$ to $\boldsymbol{\beta}$ can be quantified by optimizing the following maximization problem:

$$\mathrm{OT}^\varepsilon(\mathbf{M}, \boldsymbol{\alpha}, \boldsymbol{\beta}) = \underset{\mathbf{Q} \in \mathbf{U}(\boldsymbol{\alpha}, \boldsymbol{\beta})}{\operatorname{argmax}} \ \mathrm{Tr}(\mathbf{Q}^\top \mathbf{M}) + \varepsilon H(\mathbf{Q}), \tag{2}$$

where $\varepsilon > 0$ and $H(\mathbf{Q}) = -\sum_{ij} \mathbf{Q}_{ij} \log \mathbf{Q}_{ij}$ is the entropic regularization term. The optimal $\mathbf{Q}^*$ has been shown to be unique with the form $\mathbf{Q}^* = \mathrm{Diag}(\mathbf{u}) \exp(\mathbf{M}/\varepsilon) \mathrm{Diag}(\mathbf{v})$, where $\mathbf{u}$ and $\mathbf{v}$ can be solved by Sinkhorn's Algorithm [10].

Since $\mathbf{Q}^*$ satisfies the mass constraint (1) $\mathbf{U}(\boldsymbol{\alpha}, \boldsymbol{\beta})$ strictly, the objective $\mathrm{OT}^\varepsilon(\mathbf{M}, \boldsymbol{\alpha}, \boldsymbol{\beta})$ is not suitable for partial displacement problem. In light of this, Unbalanced OT is proposed to relax the conservation of marginal constraints by allowing the system to use soft penalties [9]. The unbalanced OT is formulated as

$$\mathrm{UOT}^{\varepsilon, \kappa}(\mathbf{M}, \boldsymbol{\alpha}, \boldsymbol{\beta}) = \underset{\mathbf{Q} \in \mathbb{R}_+}{\operatorname{argmax}} \mathrm{Tr}(\mathbf{Q}^\top \mathbf{M}) + \varepsilon H(\mathbf{Q}) - \kappa \big( D_{\mathrm{KL}}(\mathbf{Q}\mathbf{1}_c \| \boldsymbol{\alpha}) + D_{\mathrm{KL}}(\mathbf{Q}^\top \mathbf{1}_r \| \boldsymbol{\beta}) \big), \tag{3}$$

where $D_{\mathrm{KL}}$ is Kullback-Leibler Divergence. Then the optimization problem (3) can be solved by generalized Sinkhorn's algorithm [9].

## 3.2 Inter-domain Partial Alignment for Common Class Detection

To extract common knowledge, we propose an Unbalanced OT-based Common Class Detection (CCD) method, which considers a partial mapping problem between target samples and source prototypes from the perspective of cross domain. Note that our method uses prototypes to encourage a more global-aware alignment, which discovers intrinsic differences among common and target-private samples by exploiting the statistical information of the assignment matrix obtained from the optimal transport. Since UniDA allows private classes in both source and target domains, partial alignment should be considered to avoid misalignment between target-private samples and source classes. Hence we formulate the problem mapping target features to the source prototypes $\mathbf{C}_s = [\mathbf{c}_1^s, \cdots, \mathbf{c}_{|\mathcal{C}_s|}^s]^\top$ with unbalanced OT objective to obtain the optimal assignment matrix $\mathbf{Q}^{st}$, i.e.

$$\mathbf{Q}^{st} = \mathrm{UOT}^{\varepsilon, \kappa}(\mathbf{S}^{st}, \frac{1}{B}\mathbf{1}_B, \frac{1}{|\mathcal{C}_s|}\mathbf{1}_{|\mathcal{C}_s|}), \tag{4}$$

where $\mathbf{S}^{st} = \mathbf{Z}^t \mathbf{C}_s^\top$ refers to cosine similarity between target samples and source prototypes, and the mini-batch target $\ell_2$-normalized features $\mathbf{Z}^t = [\mathbf{z}_1^t, \cdots, \mathbf{z}_B^t]^\top$ with batch-size $B$ are extracted by feature extractor $f$, i.e. $\mathbf{z}_i^t = f(\mathbf{x}_i^t) / \|f(\mathbf{x}_i^t)\|_2$. Here we use the same prototype-based classifier in [32] to learn source prototypes. Note that the private samples will be assigned with relatively low weights in $\mathbf{Q}^{st}$. Based on this observation, our CCD selects target samples with top confidence as common classes.

Firstly, the assignment matrix is normalized as $\bar{\mathbf{Q}}^{st} = \mathbf{Q}^{st} / \sum \mathbf{Q}^{st}$. Then we discover statistical property from normalized $\bar{\mathbf{Q}}^{st}$ and generate scores from the statistical perspectives of both target samples and source prototypes. For the $i$-th row in $\bar{\mathbf{Q}}^{st}$, it can be seen as a vanilla prediction probability vector and we can get pseudo-label $\hat{y}_i^t$ by argmax operation. And we assign target samples confidence score $w_i^t$ with the maximum value of the $i$-th row in $\bar{\mathbf{Q}}^{st}$, i.e.

$$w_i^t = \max(\{\bar{\mathbf{Q}}_{i,1}^{st}, \bar{\mathbf{Q}}_{i,2}^{st}, \cdots, \bar{\mathbf{Q}}_{i,|\mathcal{C}_s|}^{st}\}). \tag{5}$$

A higher score $w_i^t$ means $\mathbf{z}_i^t$ is relatively closer to a source prototype than any other samples and is more likely from a common class. Meanwhile, to select target common samples with top confidence, we also evaluate source prototypes confidence score $w_j^s$ with the sum of the $j$-th column, i.e.

$$w_j^s = \sum_{i=1}^{B} \bar{\mathbf{Q}}_{i,j}^{st}. \tag{6}$$

Analogously, a higher score $w_j^s$ means $\mathbf{c}_j^s$ is more likely a common source prototype, which is assigned to target samples more frequently. Then the common samples are detected by statistics mean, i.e.

$$\delta_i = \begin{cases} 1, & w_i^t \geq \frac{1}{B} \text{ and } w_{\hat{y}_i^t}^s \geq \frac{1}{|\mathcal{C}_s|} \\ 0, & \text{otherwise} \end{cases}, \tag{7}$$

where $\delta_i = 1$ indicates the sample $\mathbf{x}_i^t$ is detected as common sample with top confidence, which can be assigned with pseudo label $\hat{y}_i^t$ by argmax operation. We can use pseudo labels of selected target samples to compute inter-domain common class detection loss by standard cross-entropy loss, i.e.

$$\mathcal{L}_{CCD} = \frac{\sum_{i=1}^B \delta_i \cdot \mathcal{L}_{CE}(\mathbf{z}_i^t, \hat{y}_i^t)}{\sum_{i=1}^B \delta_i}, \tag{8}$$

To ensure that statistics mean can discover the intrinsic difference between common and private classes, we need to guarantee that the input features for partial mapping should be sampled from the target domain distribution with enough sampling density. Therefore, we implement a FIFO memory queue [19] to save previous target features for filling mini-batch features, which avoids the limitation led by the lack of statistical property for target samples.

**Adaptive filling for unbalanced proportion of positive and negative.** Unfortunately, using statistics mean as a threshold in Eq. (7) may misclassify some common samples as private class in some extremely unbalanced cases, such as there are few private samples indeed. Essentially, the assignment weight is determined by the relative distance between samples and prototypes, and private samples with low similarity set negative examples for the others. To automatically detect target samples, we design an adaptive filling mechanism for positive and negative unbalanced proportions. Our strategy is to fill the gap of unbalanced proportion to be balanced like Fig. 2 depicted. Firstly, samples with high similarity larger than a rough boundary $\gamma$ are defined as positive and the others are negative. Then we implement adaptive filling to be balanced when the proportion of positive or negative samples exceeds $50\%$. For negative filling, we synthesize fake private feature $\mathbf{z}_i^t$ by mixing up target feature $\mathbf{z}_i^t$ and its farthest source prototypes evenly, i.e. $\hat{\mathbf{z}}_i^n = \frac{1}{2}\left(\mathbf{z}_i^t + \operatorname{argmin}_{\mathbf{c}_k^s}(\mathbf{z}_i^t \mathbf{c}_k^{s\top})\right)$.

For positive filling, we first do the CCD without adaptive filling by Eq.(7) to obtain confident features and then reuse these filtered confident features for positive filling. Note that the filling samples are randomly sampled from mix-up negatives or CCD positives and the filling size is the gap between unfilled positive and negative. After adaptive filling for balanced proportion, our CCD is adaptive for detecting confident common samples.

**Adaptive update for marginal probability vector.** The marginal probability vector $\boldsymbol{\beta}$ in Objective (4) can be interpreted as a weight budget for source prototypes to be mapped with target samples. However, the source domain may own source private class, which is unreasonable to set the weight budget equally for each source prototype, i.e. $\boldsymbol{\beta} = \frac{1}{|\mathcal{C}_s|}\mathbf{1}_{|\mathcal{C}_s|}$. Therefore, we use the $\bar{\mathbf{Q}}^{st}$ obtained in the last iteration to update $\boldsymbol{\beta}$ with the moving average for adapting to the real source distribution , i.e.

$$\boldsymbol{\beta}^{(t+1)} = \mu\boldsymbol{\beta}^{(t)} + (1-\mu)\widetilde{\boldsymbol{\beta}}^{(t)}, \quad \text{and} \quad \boldsymbol{\beta}^{(0)} = \frac{1}{|\mathcal{C}_s|}\mathbf{1}_{|\mathcal{C}_s|}, \tag{9}$$

where $\widetilde{\boldsymbol{\beta}}^{(t)}$ is the sum of column in $\bar{\mathbf{Q}}^{st}$ solved in the $\boldsymbol{\beta}^{(t)}$ case.

### 3.3 Intra-domain Representation Learning for Private Class Discovery

To exploit target global and local structure and feature representation, we propose an OT-based Private Class Discovery (PCD) clustering technique that considers a full mapping problem between target sample features and target prototypes. Especially, our PCD avoids reliance on the number of target classes and encourages global discrimination of clusters as well as local consistency of samples.

To solve this, we pre-define $K$ learnable target prototypes $\mathbf{C}_t$ and initialize them randomly, where larger $K$ can be determined by the larger number of target samples. Then we assign target features $\bar{\mathbf{Z}}^t$ with the prototypes $\mathbf{C}_t$ by solving the optimal transport optimization problem:

$$\mathbf{Q}^{tt} = \mathrm{OT}^\varepsilon(\mathbf{S}^{tt}, \frac{1}{K}\mathbf{1}_K, \frac{1}{2B}\mathbf{1}_{2B}), \tag{10}$$

where $\mathbf{S}^{tt} = \bar{\mathbf{Z}}^t \mathbf{C}_t^\top$ refers to cosine similarity between target samples and target prototypes, and $\bar{\mathbf{Z}}^t$ is the concatenation of the mini-batch target features $\mathbf{Z}^t$ and their nearest neighbor features. The marginal constraint of uniform distribution enforces that on average each prototype is selected at least $2B/K$ times in the batch.

The solution $\mathbf{Q}^{tt}$ satisfies the constraint strictly, i.e. the sum of each row equals to $1/K$. To obtain prototype assignment for each target sample, the soft pseudo-label matrix $\tilde{\mathbf{Q}}^{tt} = \mathbf{Q}^{tt} \times B$ ensures the $i$-th row $\mathbf{q}_i^{tt}$ is a probability vector. To encourage global discrimination of clusters, the learnable prototypes $\mathbf{C}_t$ and the feature extractor $f$ are optimized by minimizing $\mathcal{L}_{global}$, i.e.

$$\mathcal{L}_{global} = \frac{1}{B} \sum_{i=1}^{B} \ell(\tilde{\mathbf{q}}_i^{tt}, \mathbf{z}_i^t), \tag{11}$$

where the cross-entropy loss $\ell(\mathbf{q}_i, \mathbf{z}_i)$ is presented as

$$\ell(\mathbf{q}_i, \mathbf{z}_i) = -\sum_{k=1}^{K} q_{i,k} \log p_{i,k}, \quad \text{where} \quad p_{i,k} = \frac{\exp\left(\mathbf{z}_i^T \mathbf{c}_k / \tau\right)}{\sum_{k'=1}^{K} \exp\left(\mathbf{z}_i^T \mathbf{c}_{k'} / \tau\right)}. \tag{12}$$

To encourage local consistency of samples, we swap prediction between anchor feature $\mathbf{z}_i^t$ and its nearest neighbor feature $\tilde{\mathbf{z}}_i^t$ retrieved from the memory queue. Note that instead of swapped prediction between the different views of anchor image in SwAV [6], our swapped $\mathcal{L}_{local}$ is based on anchor-neighbor swapping, i.e.

$$\mathcal{L}_{local} = \frac{1}{2B} \sum_{i=1}^{B} \left[ \ell(\tilde{\mathbf{q}}_{B+i}^{tt}, \mathbf{z}_i^t) + \ell(\tilde{\mathbf{q}}_i^{tt}, \tilde{\mathbf{z}}_i^t) \right]. \tag{13}$$

Eventually, to encourage both global discrimination of clusters and local consistency of samples, our $\mathcal{L}_{PCD}$ is presented as

$$\mathcal{L}_{PCD} = \frac{1}{2}(\mathcal{L}_{global} + \mathcal{L}_{local}). \tag{14}$$

Similar to Sec.3.2, since OT encourages a more deliberative mapping under marginal constraint, we also need to fill input features with the previous target features in memory queue before solving the optimization problem (10).

### 3.4 A Unified Framework for UniDA by Optimal Transport

In UniDA, the partial alignment for common class detection and representation learning for private class discovery are based on UOT and OT models, respectively. Therefore, we can summarize the above two models in a unified framework, i.e. UniOT. Considering cross-entropy loss $\mathcal{L}_{cls}$ on source samples, Common Class Discovery loss $\mathcal{L}_{CCD}$ and Private Class Discovery loss $\mathcal{L}_{PCD}$, our overall objective can be expressed as

$$\mathcal{L}_{overall} = \mathcal{L}_{cls} + \lambda(\mathcal{L}_{CCD} + \mathcal{L}_{PCD}). \tag{15}$$

In the testing phase, only feature extractor $f$ and prototype-based classifier $\mathbf{C}_s$ are preserved. We solve the optimization problem (4) for all the concatenated target features and adaptive filled features. Then compute $w_j^t$ in Eq. 5 for each sample. For those samples who satisfy $w_j^t \geq \frac{1}{n}$ are assigned with the nearest source class, where $n$ is the sum of the total size of target samples $n_t$ and adaptive filling size. Otherwise, the samples are marked as *unknown*.

## 4 Experiments

### 4.1 Experimental Settings

**Dataset.** Our method will be validated in 4 popular datasets in Domain Adaptation. **Office** [31] contains 31 categories and about 4K images in 3 domains: Amazon(**A**), DSLR(**D**) and Webcam(**W**). **Office-Home** [37] contains 65 categories and about 15K images in 4 domains: Artistic images(**Ar**), Clip-Art images(**Cl**), Product images(**Pr**) and Real-World images(**Rw**). **VisDA** [30] is a large dataset

which contains 12 categories in source domain with 15K synthetic images and target domain with 5K real-world images. **DomainNet** [29] is the largest domain adaptation dataset which contains 345 categories and about 0.6 million in 6 domains but we only use 3 domain: Painting (**P**), Real (**R**), and Sketch (**S**) like [17]. Note that we use **A2W** to denote that transfer task from Amazon to DSLR. We follow the dataset split in [17] to conduct experiments.

**Evaluation metric.** For UniDA, the trade-off between the accuracy of common and private classes is important in evaluating performance. Thus, we use the H-score [17] to calculate the harmonic mean of the instance accuracy on common class $a_{\mathcal{C}}$ and accuracy on the single private class $a_{\bar{\mathcal{C}}_t}$. However, H-score treats all private samples as a single class. For target-private class discovery purposes, we further introduce $H^3$-score, as the harmonic mean of accuracy on common class, accuracy on unknown class and Normalized Mutual Information (NMI) for target-private clusters, i.e.

$$H^3\text{-score} = \frac{3}{1/a_{\mathcal{C}} + 1/a_{\bar{\mathcal{C}}_t} + 1/\text{NMI}}, \tag{16}$$

where NMI is a well-known measure for the quality of clustering, and NMI is obtained by K-Means with the ground truth of the number of private classes in the inference stage.

## 4.2 Experimental details

Our implementation of optimal transport solver is based on POT [15]. We employ pretrained Res-Net-50 [20] on ImageNet [12] as our initial backbone. The feature extractor consists of backbone and projection layers which are the same as [6] but BatchNorm layer is removed. We adopt the same optimizer details as [23]. The batch size is set to 36 for both source and target domains. For all experiments, the initial learning rate is set to $1 \times 10^{-2}$ for all new layers and $1 \times 10^{-3}$ for pretrained backbone. The total training steps are set to be 10K for all datasets. For the pre-defined number of target prototypes, a larger size of target domain indicates a larger $K$. Therefore, we empirically set $K = 50$ for Office, $K = 150$ for Office-Home, $K = 500$ for VisDA, $K = 1000$ for DomainNet. We default $\gamma = 0.7$, $\mu = 0.7$, $\tau = 0.1$, $\varepsilon = 0.01$, $\kappa = 0.5$ and $\lambda = 0.1$ for all datasets. We set the size of memory queue 2K for Office and Office-Home, 10K for VisDA and DomainNet.

All experiments are implemented on a GPU of NVIDIA TITAN V with 12GB. Each experiment takes about 2.5 hours. Our code is available at: `https://github.com/changwxx/UniOT-for-UniDA`.

## 4.3 Results

**Comparison with state-of-the-arts.** Tab. 1 and 2 show the H-score results for Office, Office-Home, VisDA and DomainNet. Our UniOT achieves the best performance on all benchmarks. H-score on non-UniDA methods are reported from [17], and UniOT methods are reported from original papers accordingly, except for DANCE [32] since they did not report H-score and we reproduce it with their released code for fair comparison, marked as ‡. In particular, with respect to $H^3$-score in Tab. 3, our UniOT surpasses other methods by 7% and 9% in Office and Office-Home datasets respectively, which demonstrates that our UniOT achieves balanced performance for both common class detection and private class discovery.

| | **Office** | | | | | | | **DomainNet** | | | | | | |
|---|---|---|---|---|---|---|---|---|---|---|---|---|---|---|
| | A2D | A2W | D2A | D2W | W2A | W2D | Avg | P2R | R2P | P2S | S2P | R2S | S2R | Avg |
| ResNet[20] | 49.78 | 47.92 | 48.48 | 54.94 | 48.96 | 55.60 | 50.94 | 30.06 | 28.34 | 26.95 | 26.95 | 26.89 | 29.74 | 28.15 |
| DANN[18] | 50.18 | 48.82 | 47.69 | 52.73 | 49.33 | 54.87 | 50.60 | 31.18 | 29.33 | 27.84 | 27.84 | 27.77 | 30.84 | 29.13 |
| RTN[25] | 50.18 | 50.21 | 47.65 | 54.68 | 49.28 | 55.24 | 51.21 | 32.27 | 30.29 | 28.71 | 28.71 | 28.63 | 31.90 | 30.08 |
| IWAN[40] | 50.64 | 50.13 | 49.65 | 54.06 | 49.79 | 55.44 | 51.62 | 35.38 | 33.02 | 31.15 | 31.15 | 31.06 | 34.94 | 32.78 |
| PADA[4] | 50.00 | 49.65 | 42.87 | 52.62 | 49.17 | 55.60 | 49.98 | 28.92 | 27.32 | 26.03 | 26.03 | 25.97 | 28.62 | 27.15 |
| ATI[3] | 50.48 | 48.58 | 48.48 | 55.01 | 48.98 | 55.45 | 51.16 | 32.59 | 30.57 | 28.96 | 28.96 | 28.89 | 32.21 | 30.36 |
| OSBP[33] | 51.14 | 50.23 | 49.75 | 55.53 | 50.16 | 57.20 | 52.34 | 33.60 | 33.03 | 30.55 | 30.53 | 30.61 | 33.65 | 32.00 |
| UAN[39] | 59.68 | 58.61 | 60.11 | 70.62 | 60.34 | 71.42 | 63.46 | 41.85 | 43.59 | 39.06 | 38.95 | 38.73 | 43.69 | 40.98 |
| CMU[17] | 68.11 | 67.33 | 71.42 | 79.32 | 72.23 | 80.42 | 73.14 | 50.78 | **52.16** | 45.12 | 44.82 | 45.64 | 50.97 | 48.25 |
| DANCE‡[32] | 72.64 | 62.43 | 63.27 | 76.29 | 57.37 | 82.79 | 66.62 | - | - | - | - | - | - | - |
| DCC[23] | **88.50** | 78.54 | 70.18 | 79.29 | 75.87 | 88.58 | 80.16 | 56.90 | 50.25 | 43.66 | 44.92 | 43.31 | 56.15 | 49.20 |
| TNT[8] | 85.70 | 80.40 | 83.80 | 92.00 | 79.10 | 91.20 | 85.37 | - | - | - | - | - | - | - |
| **UniOT** | 86.97 | **88.48** | **88.35** | **98.83** | **87.60** | **96.57** | **91.13** | **59.30** | 47.79 | **51.79** | **46.81** | **48.32** | **58.25** | **52.04** |

Table 1: H-score(%) on **Office** and **DomainNet**.

| | Office-Home | | | | | | | | | | | | | VisDA |
|---|---|---|---|---|---|---|---|---|---|---|---|---|---|---|
| | Ar2Cl | Ar2Pr | Ar2Rw | Cl2Ar | Cl2Pr | Cl2Rw | Pr2Ar | Pr2Cl | Pr2Rw | Rw2Ar | Rw2Cl | Rw2Pr | Avg | |
| ResNet[20] | 44.65 | 48.04 | 50.13 | 46.64 | 46.91 | 48.96 | 47.47 | 43.17 | 50.23 | 48.45 | 44.76 | 48.43 | 47.32 | 25.44 |
| DANN[18] | 42.36 | 48.02 | 48.87 | 45.48 | 46.47 | 48.37 | 45.75 | 42.55 | 48.70 | 47.61 | 42.67 | 47.40 | 46.19 | 25.65 |
| RTN[25] | 38.41 | 44.65 | 45.70 | 42.64 | 44.06 | 45.48 | 42.56 | 36.79 | 45.50 | 44.56 | 39.79 | 44.53 | 42.89 | 26.02 |
| IWAN[40] | 40.54 | 46.96 | 47.78 | 44.97 | 45.06 | 47.59 | 45.81 | 41.43 | 47.55 | 46.29 | 42.49 | 46.54 | 45.25 | 27.64 |
| PADA[4] | 34.13 | 41.89 | 44.08 | 40.56 | 41.52 | 43.96 | 37.04 | 32.64 | 44.17 | 43.06 | 35.84 | 43.35 | 40.19 | 23.05 |
| ATI[3] | 39.88 | 45.77 | 46.63 | 44.13 | 44.39 | 46.63 | 44.73 | 41.20 | 46.59 | 45.05 | 41.78 | 45.45 | 44.35 | 26.34 |
| OSBP[33] | 39.59 | 45.09 | 46.17 | 45.70 | 45.24 | 46.75 | 45.26 | 40.54 | 45.75 | 45.08 | 41.64 | 46.90 | 44.48 | 27.31 |
| UAN[39] | 51.64 | 51.70 | 54.30 | 61.74 | 57.63 | 61.86 | 50.38 | 47.62 | 61.46 | 62.87 | 52.61 | 65.19 | 56.58 | 30.47 |
| CMU[17] | 56.02 | 56.93 | 59.15 | 66.95 | 64.27 | 67.82 | 54.72 | 51.09 | 66.39 | 68.24 | 57.89 | 69.73 | 61.60 | 34.64 |
| DANCE‡[32] | 26.67 | 11.27 | 18.03 | 33.17 | 12.50 | 14.33 | 41.56 | 39.92 | 33.34 | 16.31 | 27.12 | 25.86 | 25.01 | - |
| DCC[23] | 57.97 | 54.05 | 58.01 | **74.64** | 70.62 | 77.52 | 64.34 | **73.60** | 74.94 | **80.96** | **75.12** | 80.38 | 70.18 | 43.02 |
| TNT[8] | 61.90 | 74.60 | 80.20 | 73.50 | 71.40 | 79.60 | 74.20 | 69.50 | 82.70 | 77.30 | 70.10 | 81.20 | 74.70 | 55.30 |
| **UniOT** | **67.27** | **80.54** | **86.03** | 73.51 | **77.33** | **84.28** | **75.54** | 63.33 | **85.99** | 77.77 | 65.37 | **81.92** | **76.57** | **57.32** |

Table 2: H-score(%) on **Office-Home** and **VisDA**.

| | Office | | | | | | | | Office-Home | | | | | | | | | | | | |
|---|---|---|---|---|---|---|---|---|---|---|---|---|---|---|---|---|---|---|---|---|---|
| | A2D | A2W | D2A | D2W | W2A | W2D | Avg | | Ar2Cl | Ar2Pr | Ar2Rw | Cl2Ar | Cl2Pr | Cl2Rw | Pr2Ar | Pr2Cl | Pr2Rw | Rw2Ar | Rw2Cl | Rw2Pr | Avg |
| ResNet[20] | 53.90 | 51.79 | 46.81 | 59.15 | 46.54 | 61.32 | 53.25 | | 41.42 | 50.88 | 49.56 | 43.55 | 46.98 | 46.62 | 45.65 | 40.38 | 50.08 | 46.57 | 41.70 | 50.84 | 46.18 |
| UAN[39] | 66.15 | 64.20 | 57.90 | 72.63 | 57.93 | 75.73 | 65.76 | | 48.86 | 57.19 | 58.35 | 58.80 | 61.42 | 62.80 | 51.67 | 46.11 | 63.24 | 60.69 | 49.40 | 67.62 | 57.18 |
| DANCE[32] | 73.19 | 68.53 | 67.88 | 81.09 | 65.61 | 85.70 | 73.67 | | 40.92 | 40.95 | 45.84 | 29.73 | 20.26 | 36.97 | 52.63 | 48.23 | 50.13 | 22.78 | 44.89 | 58.29 | 40.97 |
| DCC[23] | **84.47** | 74.80 | 63.54 | 87.09 | 69.58 | 71.55 | 75.17 | | 55.64 | 78.21 | 78.18 | 44.64 | 33.77 | 69.96 | 63.77 | 53.81 | 65.10 | 63.17 | 53.58 | **80.09** | 61.66 |
| **UniOT** | 83.69 | **85.28** | **71.46** | **91.24** | **70.93** | **90.84** | **82.24** | | **60.11** | **78.72** | **79.53** | **65.83** | **75.32** | **76.83** | **68.21** | **56.83** | **80.55** | **69.62** | **58.74** | 79.84 | **70.84** |

Table 3: H$^3$-score(%) on **Office** and **Office-Home**

**Evaluation of the effectiveness of the proposed CCD and PCD.** To evaluate the contribution of $\mathcal{L}_{CCD}$, $\mathcal{L}_{global}$ and $\mathcal{L}_{local}$, we train the model with different combination of each component. As shown in Tab.4, the combination of $\mathcal{L}_{global}$ and $\mathcal{L}_{local}$, i.e. $\mathcal{L}_{PCD}$, brings significant contribution since target representation is crucial for distinguishing common and private samples. Especially, $\mathcal{L}_{global}$ contributes more than $\mathcal{L}_{local}$, which demonstrates the benefit of global discrimination of clusters for representation learning. Besides, we also conduct experiments for CCD without adaptive filling and denote the loss as $\mathcal{L}_{CCD}^{\dagger}$, which verifies the effectiveness of adaptive filling design.

| | | | | H-score | | | | | | H$^3$-score | | | | | |
|---|---|---|---|---|---|---|---|---|---|---|---|---|---|---|---|
| | | | | Office | | | Office-Home | | | Office | | | Office-Home | | |
| $\mathcal{L}_{CCD}$ | $\mathcal{L}_{CCD}^{\dagger}$ | $\mathcal{L}_{global}$ | $\mathcal{L}_{local}$ | A2W | D2A | Avg (6 tasks) | Ar2Pr | Cl2Rw | Avg (12 tasks) | A2W | D2A | Avg (6 tasks) | Ar2Pr | Cl2Rw | Avg (12 tasks) |
| ✓ | | | | 77.98 | 87.79 | 83.57 | 71.21 | 74.24 | 69.73 | 69.95 | 67.44 | 72.94 | 64.84 | 59.30 | 58.18 |
| ✓ | | | ✓ | 86.81 | 86.36 | 88.44 | 73.53 | 76.49 | 71.40 | 82.95 | 67.88 | 81.18 | 73.14 | 69.37 | 65.34 |
| ✓ | | ✓ | | 87.71 | 84.71 | 89.04 | 80.00 | 83.83 | 76.07 | 80.36 | 66.31 | 77.68 | 78.32 | 76.73 | 69.89 |
| | | ✓ | ✓ | 74.18 | 72.61 | 79.74 | 79.59 | 74.24 | 75.55 | 75.38 | 62.08 | 76.18 | 78.42 | 76.12 | 70.44 |
| | ✓ | ✓ | ✓ | 87.84 | **89.19** | 89.86 | 75.10 | 79.14 | 72.65 | 81.25 | 70.75 | 80.49 | 75.28 | 74.02 | 68.31 |
| ✓ | | ✓ | ✓ | **88.48** | 88.35 | **91.13** | **80.54** | **84.28** | **76.57** | **85.28** | **71.46** | **82.24** | **78.72** | **76.83** | **70.84** |

Table 4: Evaluation of the effectiveness of the proposed CCD and PCD.

**Robustness in realistic UniDA.** We compare the performance of our UniOT with DCC, DANCE and UAN under different categories split. In this analysis, we perform on A2W of Office with a fixed common label set $\mathcal{C}$ and target-private label set $\overline{\mathcal{C}}_t$, but the different size of source-private label set $\overline{\mathcal{C}}_s$. We set 10 categories for $\mathcal{C}$, 11 categories for $\overline{\mathcal{C}}_t$ and vary the number of $\overline{\mathcal{C}}_s$. Fig. 3(a) shows that our UniOT always outperforms other methods. Besides, UniOT is not sensitive to the variation of source-private classes while other methods are not stable enough. Additionally, we also analyze the behavior of UniOT under the different size of target-private label set $\overline{\mathcal{C}}_t$. We also perform on A2W of Office with a fixed common label set $\mathcal{C}$ for 10 categories and source-private label set $\overline{\mathcal{C}}_s$ for 10 categories. Fig. 3(b) shows that our UniOT still performs better, while the other methods present sensitivity to the number of target-private classes. Therefore, UniOT is more robust to the variation of target-private classes.

**Feature visualization for comparison with existing UniDA methods.** We use t-SNE [35] to visualize the learned target features for Rw2Pr of Office-Home. As shown in Fig. 4, the common samples are colored in black and the private samples are colored with other non-black colors by their ground-truth classes. Fig. 4(b) shows that UAN cannot recognize different categories among target-private samples since they treat all target-private samples as a single class. Especially, our UniOT discovers the global and local structure of target-private samples. Fig. 4(d) validates that UniOT learns a better target representation with global discrimination and local consistency of samples compared to DANCE shown in Fig. 4(c).

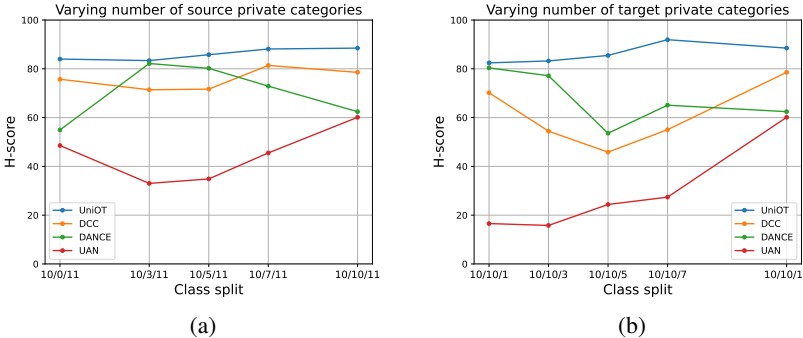

| (a) | (b) |

Figure 3: Robustness in realistic UniDA with diverse ratios of common classes. Different class splits $\mathcal{C}/\overline{\mathcal{C}}_s/\overline{\mathcal{C}}_t$ reveal diverse ratios of common classes.

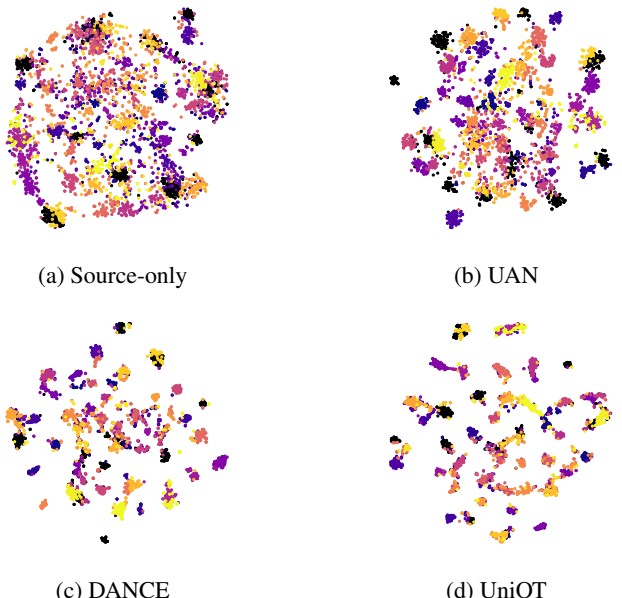

| (a) Source-only | (b) UAN |
| (c) DANCE | (d) UniOT |

Figure 4: Feature visualization of target domain on Rw2Pr of Office-Home.

## 5 Conclusion

In this paper, we have proposed to use Optimal Transport to handle common class detection and private class discovery for UniDA under a unified framework, namely UniOT. More precisely, an OT-based cross-domain partial alignment was designed to detect common class with an adaptive filling strategy to handle the diverse UniDA settings. In addition, we have proposed an OT-based target representation learning for private class discovery, which encourages both global discrimination and local consistency of samples. Experimental results on four benchmark datasets have validated the superiority of our UniOT over a wide range of state-of-the-art methods. In the future, we will modify memory model for more efficient features filling in OT.

## 6 Acknowledgement

This work was supported by the Shanghai Sailing Program (21YF1429400, 22YF1428800), Shanghai Local College Capacity Building Program (23010503100), and Shanghai Frontiers Science Center of Human-centered Artificial Intelligence (ShangHAI).

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
