# Unified Optimal Transport Framework for Universal Domain Adaptation
# (Supplementary Material)

**Wanxing Chang**[1]    **Ye Shi**[1,3*]    **Hoang Duong Tuan**[2]    **Jingya Wang**[1,3*]

[1]ShanghaiTech University    [2]University of Technology Sydney
[3]Shanghai Engineering Research Center of Intelligent Vision and Imaging

{changwx,shiye,wangjingya}@shanghaitech.edu.cn
tuan.hoang@uts.edu.au

## A   Supplement for experimental settings

### A.1   Evaluation metric

**H-score** was proposed in CMU [4], which emphasizes the importance of both abilities of UniDA methods. Inspired by F1-score, H-score is defined as the harmonic mean of the instance accuracy on common class $a_{\mathcal{C}}$ and and accuracy on the "unknown" class $a_{\bar{\mathcal{C}}_t}$ as:

$$\text{H-score} = \frac{2}{1/a_{\mathcal{C}} + 1/a_{\bar{\mathcal{C}}_t}} = 2 \cdot \frac{a_{\mathcal{C}} \cdot a_{\bar{\mathcal{C}}_t}}{a_{\mathcal{C}} + a_{\bar{\mathcal{C}}_t}}. \tag{S1}$$

H-score is high only when both the $a_{\mathcal{C}}$ and $a_{\bar{\mathcal{C}}_t}$ are high, and has applied as a fair evaluation metric in UniDA [4, 6, 2].

### A.2   Dataset split.

We follow the dataset split in [4, 10] to conduct experiments. Here we present more details about the dataset split. As we state in Section 2, we denote $\mathcal{C} = \mathcal{C}_s \cap \mathcal{C}_t$ as the common label set shared by both domains and let $\bar{\mathcal{C}}_s = \mathcal{C}_s \backslash \mathcal{C}$ and $\bar{\mathcal{C}}_t = \mathcal{C}_t \backslash \mathcal{C}$ represent label sets of source private and target private, respectively.

**Office.** UAN [10] uses the 10 classes shared by Office-31 and Caltech-256 as the common label set $\mathcal{C}$, then in alphabetical order, the next 10 classes are used as the $\bar{\mathcal{C}}_s = \mathcal{C}_s \backslash \mathcal{C}$, and the reset 11 classes are used as the $\bar{\mathcal{C}}_t = \mathcal{C}_t \backslash \mathcal{C}$. **Office-Home.** In alphabet order, UAN [10] uses the first 10 classes as $\mathcal{C}$, the next 5 classes as $\bar{\mathcal{C}}_s = \mathcal{C}_s \backslash \mathcal{C}$ and the rest 55 classes as $\bar{\mathcal{C}}_t = \mathcal{C}_t \backslash \mathcal{C}$. **VisDA.** UAN [10] uses the first 6 classes as $\mathcal{C}$, the next 3 classes as $\bar{\mathcal{C}}_s = \mathcal{C}_s \backslash \mathcal{C}$ and the rest as $\bar{\mathcal{C}}_t = \mathcal{C}_t \backslash \mathcal{C}$. **DomainNet.** CMU [4] uses the first 150 classes as $\mathcal{C}$, the next 50 classes as $\bar{\mathcal{C}}_s = \mathcal{C}_s \backslash \mathcal{C}$ and the rest as $\bar{\mathcal{C}}_t = \mathcal{C}_t \backslash \mathcal{C}$.

## B   Additional results

**H³-score on VisDA and DomainNet.** Tab. S1 shows the comparison of H³-score between UniOT and the state-of-the-arts for VisDA and DomainNet. Our UniOT surpasses the other methods by a

---

*Corresponding author.

36th Conference on Neural Information Processing Systems (NeurIPS 2022).

large margin, especially for VisDA. We omit $H^3$-score for TNT [2] and DCC [6] since they did not release code or hyper-parameters for these two datasets.

**H-score with standard deviation.** We report error bars after running the experiments three times for UniOT. As shown in Tab. S2, we report the averaged H-score results as well as standard deviation (std) for Office and Office-Home. We observe that the std values are generally close to zero for all transfer tasks, which demonstrates the stability of our method.

| | VisDA | DomainNet | | | | | | |
|---|---|---|---|---|---|---|---|---|
| | | P2R | R2P | P2S | S2P | R2S | S2R | Avg |
| ResNet[5] | 34.55 | 36.21 | 33.15 | 30.84 | 31.50 | 30.76 | 35.53 | 33.00 |
| UAN[10] | 30.05 | 38.39 | 37.62 | 39.65 | 34.84 | 33.17 | 49.54 | 38.87 |
| DANCE[8] | 6.33 | 42.45 | 42.91 | **50.03** | 45.35 | 42.50 | 44.09 | 44.56 |
| **UniOT** | **47.23** | **46.85** | **51.75** | 47.00 | **47.98** | **58.56** | **58.56** | **51.78** |

Table S1: $H^3$-score(%) on **VisDA** and **DomainNet**.

| Office | | | | | | |
|---|---|---|---|---|---|---|
| A2D | A2W | D2A | D2W | W2A | W2D | Avg |
| 86.97±1.08 | 88.48±0.66 | 88.35±0.56 | 98.83±0.22 | 87.60±0.35 | 96.57±0.00 | 91.13±0.23 |

| Office-Home | | | | | |
|---|---|---|---|---|---|
| Ar2Cl | Ar2Pr | Ar2Rw | Cl2Ar | Cl2Pr | Cl2Rw |
| 67.27±0.19 | 80.54±0.42 | 86.03±0.27 | 73.51±0.58 | 77.33±0.64 | 84.28±0.14 |

| Pr2Ar | Pr2Cl | Pr2Rw | Rw2Ar | Rw2Cl | Rw2Pr | Avg |
|---|---|---|---|---|---|---|
| 75.54±0.45 | 63.33±0.74 | 85.99±0.19 | 77.77±0.56 | 65.37±0.25 | 81.92±0.22 | 76.57±0.17 |

Table S2: Averaged H-score with standard deviation (after three runs) for **Office** and **Office-Home**.

**Effect of Common Class Detection (CCD).** To show that our CCD can effectively detect common samples as the training progresses, we present the evolution of Recall [7] and Specificity [7] values for D2W of Office. Recall measures the fraction of common samples that are retrieved as correct common class, while specificity measures the fraction of private samples that are not retrieved. As shown in Fig. S1(a), our CCD guarantees high recall rate and low specificity rate, which verifies that our CCD can detect common samples for domain alignment and avoid misalignment for target-private samples.

**Sensitivity to hyper-parameters.** Fig. S1(b) shows the sensitivity of $\gamma$, where $\gamma$ is the rough boundary for splitting positive and negative in adaptive filling. For the cosine similarity of two $\ell_2$-normalized features, the similarity value is limited from $-1$ to $1$, where higher value indicates higher similarity. To rough split positive and negative, the boundary $\gamma$ should be high enough. We can observe that the performance is not sensitive to the different $\gamma$.

Fig. S1(c) shows the sensitivity of moving average factor $\mu$ in Eq. 9. When $\mu = 1$, the performance is not good since $\mu = 1$ means no adaptive update for the marginal probability vector, which cannot fit actual budget for source prototypes. When $\mu < 1$, the performance does boost and it is not sensitive to different $\mu$, which demonstrates the positive effects of adaptive update for the marginal probability vector.

Fig. S1(d) shows the sensitivity of the overall loss weight factor $\lambda$ in Eq. 15 , which demonstrates that our method is not sensitive to the different selection of $\lambda$ varying from $0.1$ to $0.5$.

| | H-score | | | | $H^3$-score | | | |
|---|---|---|---|---|---|---|---|---|
| | VisDA | P2S | R2P | S2P | VisDA | P2S | R2P | S2P |
| $\mathcal{L}_{CCD} + \mathcal{L}_{global} + \mathcal{L}_{SwAV}$ | 44.57 | 43.20 | 45.97 | 45.39 | 44.62 | 37.96 | 43.91 | 41.91 |
| $\mathcal{L}_{CCD} + \mathcal{L}_{SimSiam}$ | 53.49 | 44.31 | 45.43 | 45.71 | 44.65 | 38.49 | 44.09 | 42.25 |
| $\mathcal{L}_{CCD} + \mathcal{L}_{PCD}$ | **57.32** | **51.79** | **47.79** | **46.81** | **60.33** | **47.00** | **51.75** | **47.98** |

Table S3: Effect of local neighbor for representation learning.

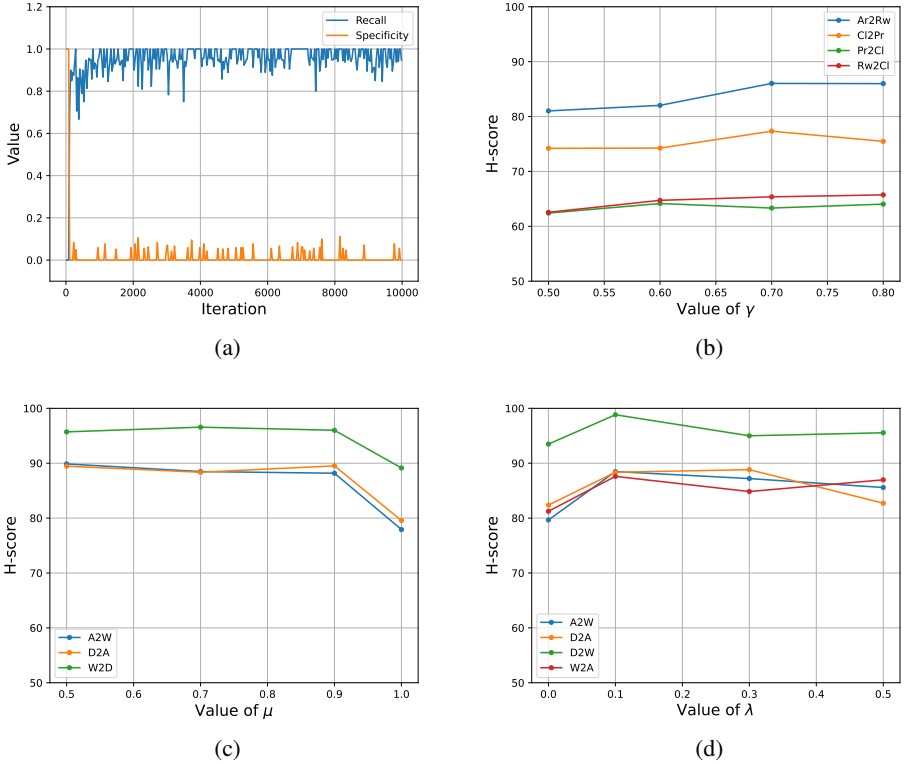

Figure S1: (a) The evolution of Recall and Specificity on D2W of Office. (b) Sensitivity to the rough boundary $\gamma$ for adaptive filling on Office-Home. (c) Sensitivity to the moving average factor $\mu$ for adaptive update of marginal probability vector on Office. (d) Sensitivity to the overall loss weight factor $\lambda$ on Office.

**Effect of local neighbor for representation learning.** To show that the local neighbor consistency does benefit the representation learning in target domain, we conduct experiments with small-batch self-supervised learning methods, such as SimSiam [3] and SwAV [1]. Such self-supervised learning methods encourage the consistency between two augmentations of one image. We conduct the experiments on VisDA and three transfer tasks on DomainNet. For SimSiam, we replace $\mathcal{L}_{PCD}$ with the SimSiam loss $\mathcal{L}_{SimSiam}$. For SwAV, we replace $\mathcal{L}_{local}$ with the SwAV loss $\mathcal{L}_{SwAV}$. Notably, Tab. S3 shows that directly combining self-supervised methods with domain adaptation barely makes a contribution. One possible interpretation is that the self-supervised methods mainly contribute to the case without any supervision but fail to benefit under the powerful source supervision. Therefore, learning from local neighbor performs better than augmentation-based self-supervised learning methods for UniDA.

**Feature visualization for the significance of Private Class Discovery.** We use t-SNE [9] to visualized the both source and target features for Rw2Pr of Office-Home for our UniOT. Notably, Office-Home is a more challenging dataset where there are 50 target-private categories but only 10 common categories. Source and target common categories are printed as colourful 0-9 digits, 5 source-common categories are printed as brown capitalized letters A-E, and the left 50 target-private categories are printed as grey triangles for simplicity. As shown in Fig. S2, we can observe that our Private Class Discovery (PCD) encourages more compact representation for common classes, which can improve accuracy for common classes, such as class-2 in the figure. Moreover, learning without PCD presents disordered representation around source-private classes, which will cause acute mis-classification for target-private samples, such as class-A, class-B, class-E printed in brown. Our PCD refines this situation significantly by encouraging self-supervision for target-private samples instead of over-reliance on source supervision.

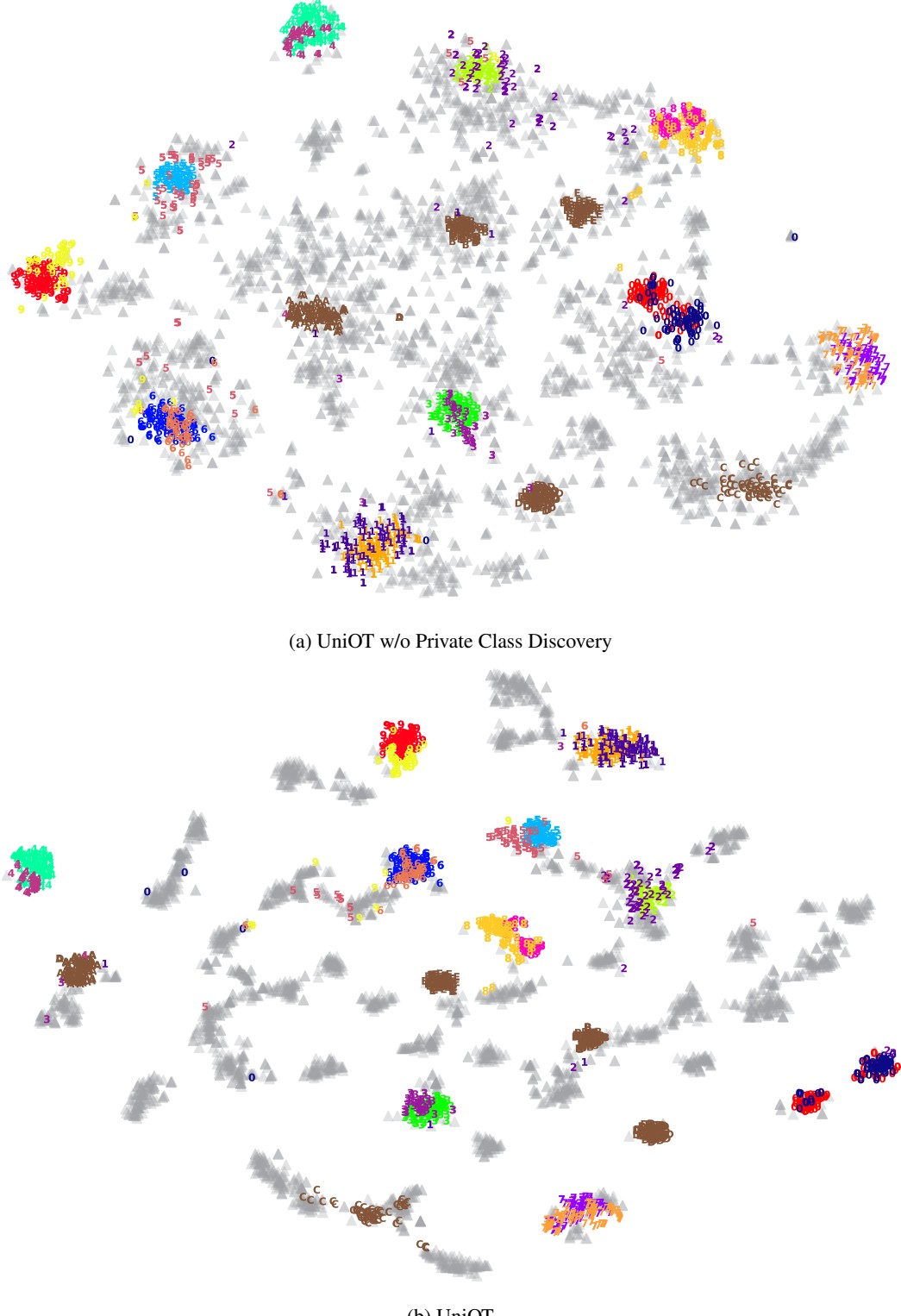

(a) UniOT w/o Private Class Discovery

(b) UniOT

Figure S2: Feature visualization for both source and target domain on Rw2Pr of Office-Home. Source-common set: {0, 1, 2, 3, 4, 5, 6, 7, 8, 9}, source-private set: {A, B, C, D, E}, target-common set: {0, 1, 2, 3, 4, 5, 6, 7, 8, 9}, target-private samples are printed as ▲. Private Class Discovery (PCD) helps UniOT learn more compact representation for target domain, which improves the accuracy of both common classes and target-private classes.

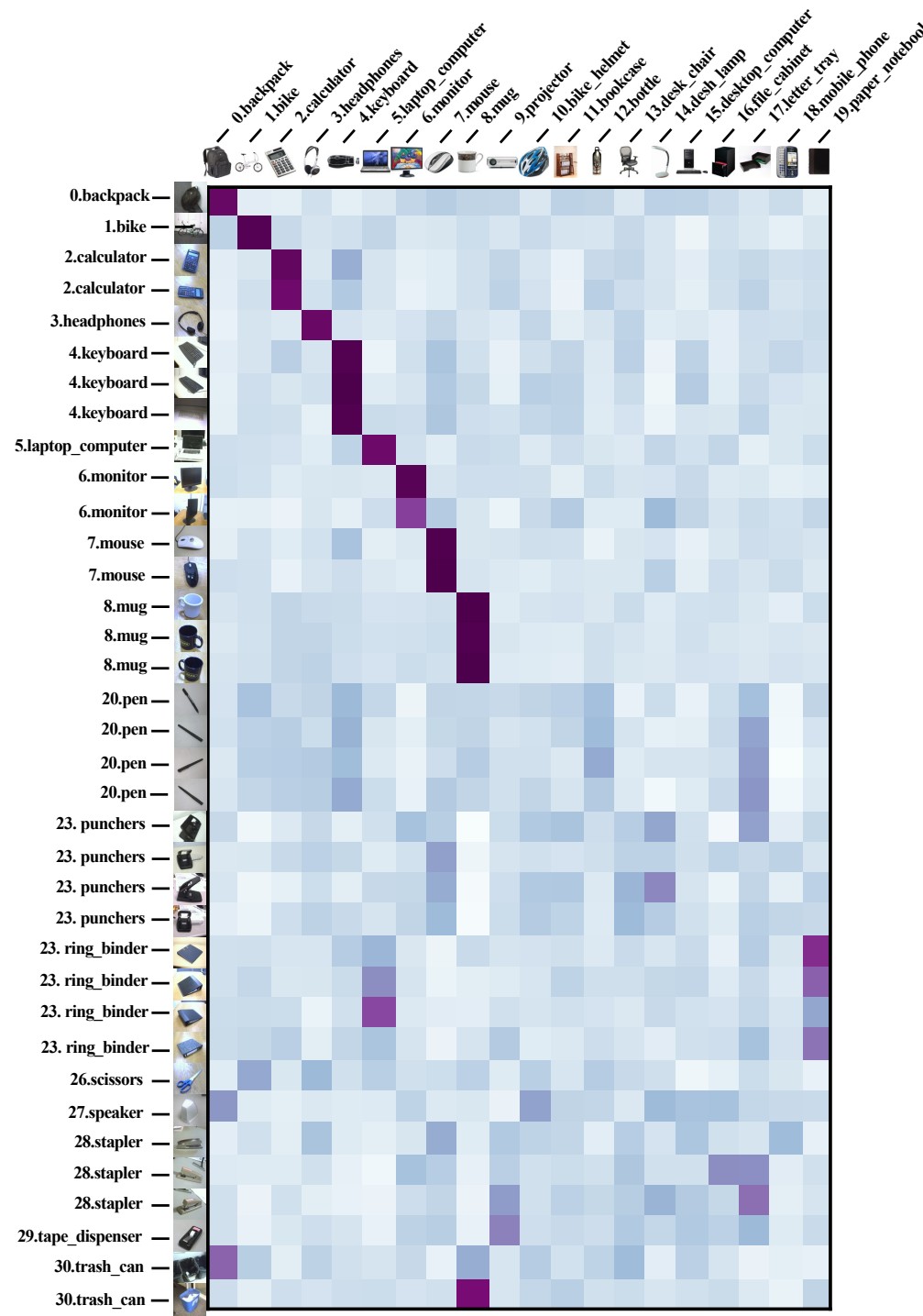

Figure S3: Qualitative illustration of similarity matrix $\mathbf{S}^{st}$ for batch target samples and source prototypes on A2W in Office. The display images for source prototypes are chosen by finding the nearest source instance of the prototype. In A2W task, 0-9 are common classes, 10-19 are source-private classes and 20-30 are target-private classes.

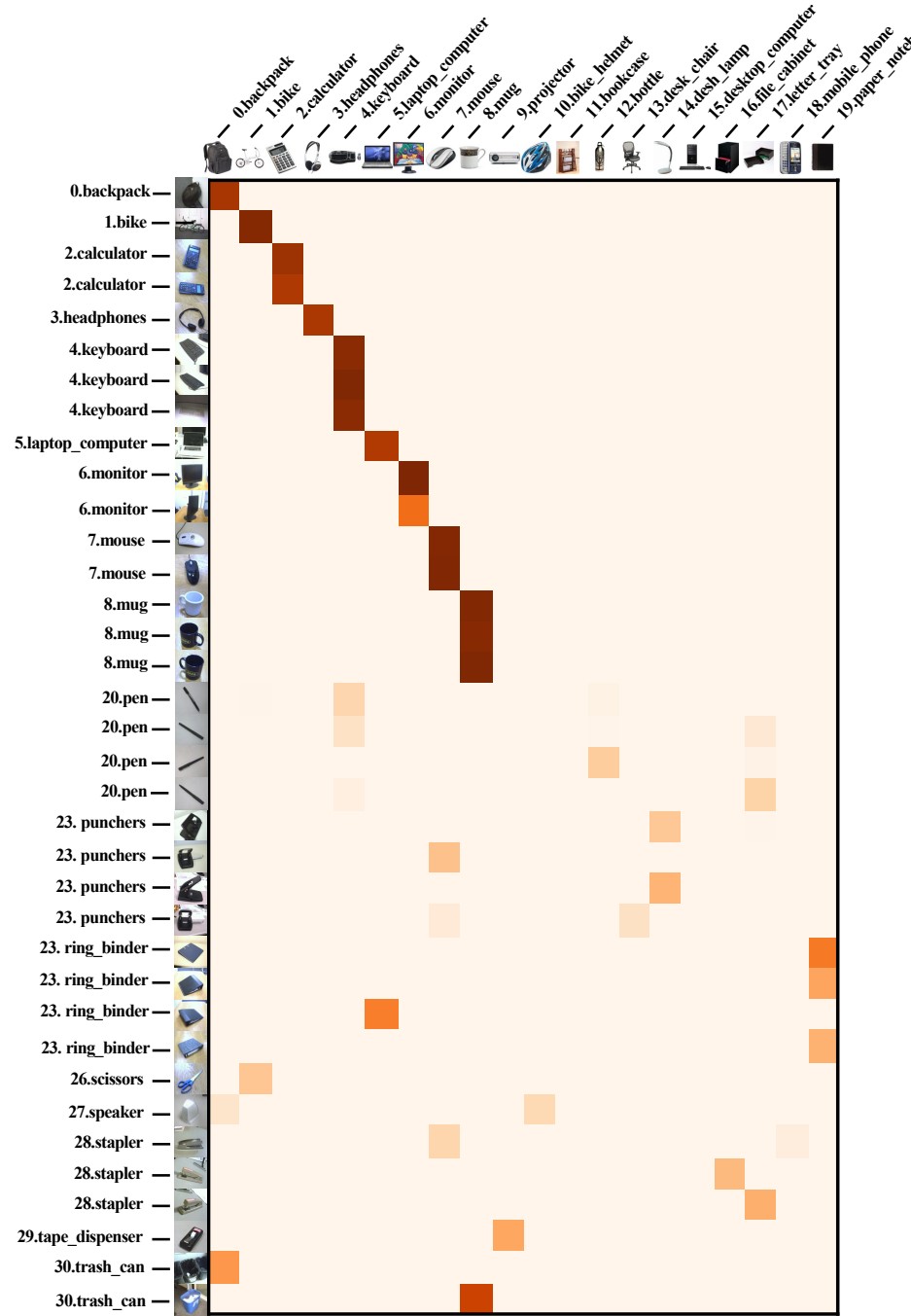

Figure S4: Qualitative illustration of UOT coupling matrix $\mathbf{Q}^{st}$ for batch target samples and source prototypes on A2W in Office. The display images for source prototypes are chosen by finding the nearest source instance of the prototype. In A2W task, 0-9 are common classes, 10-19 are source-private classes and 20-30 are target-private classes.

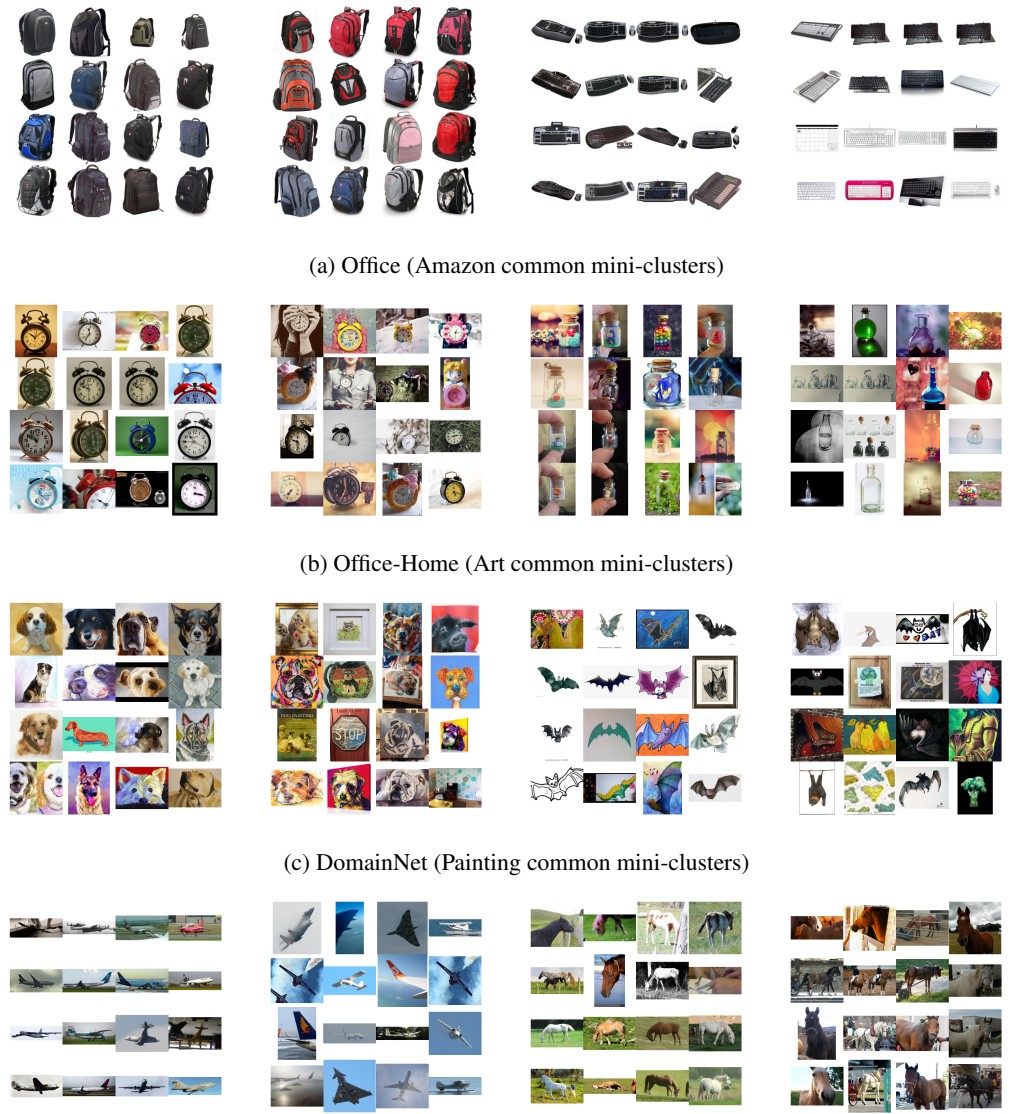

(a) Office (Amazon common mini-clusters)

(b) Office-Home (Art common mini-clusters)

(c) DomainNet (Painting common mini-clusters)

(d) VisDA (common mini-clusters)

Figure S5: Examples of common mini-clusters. Top-16 nearest neighbors of common target prototypes are presented and we show 2 mini-clusters of 2 common classes for each dataset.

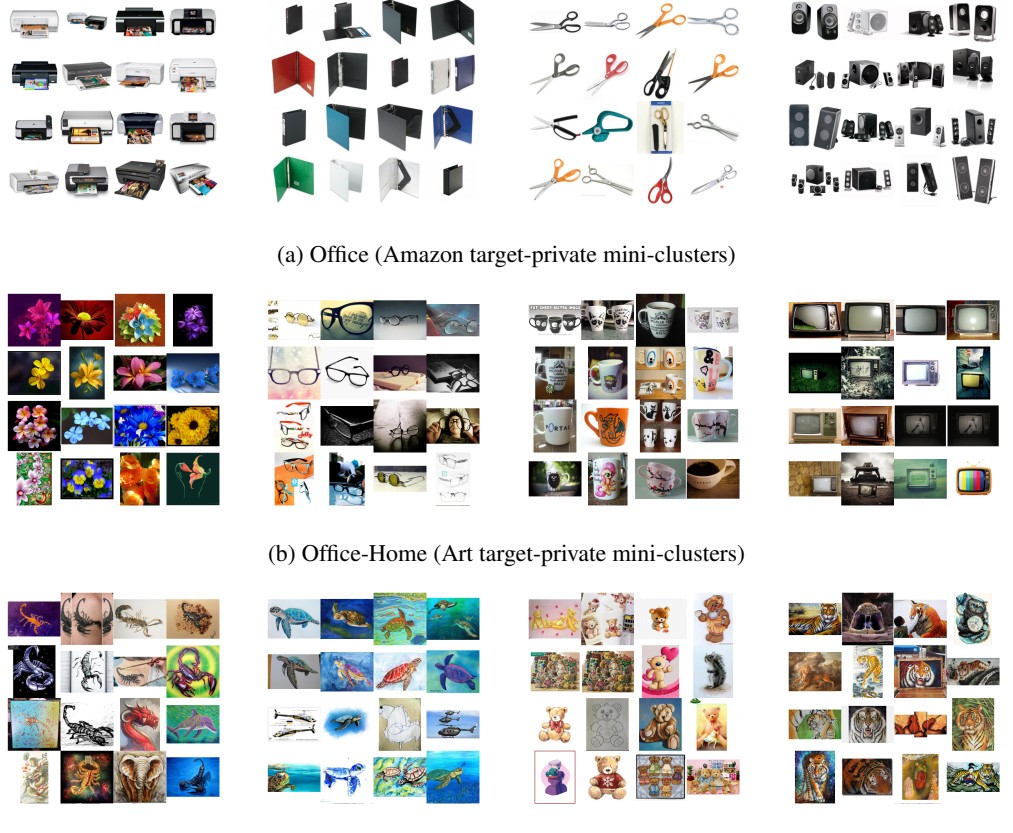

(a) Office (Amazon target-private mini-clusters)

(b) Office-Home (Art target-private mini-clusters)

(c) DomainNet (Painting target-private mini-clusters)

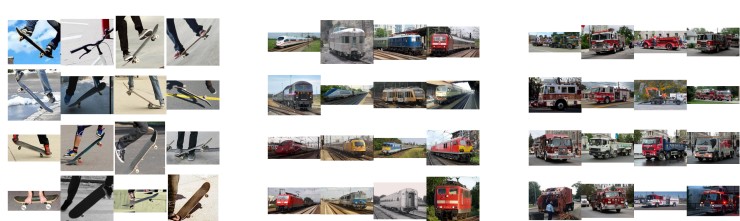

(d) VisDA (target-private mini-clusters)

Figure S6: Examples of target-private mini-clusters. Top-16 nearest neighbors of target-private prototypes are presented and we show one mini-clusters of 4 target-private classes for each dataset (except for VisDA which owns 3 target-private classes only).

**Qualitative illustration for cross-domain mapping matrix.** To better illustrate the process of mapping from target samples to source prototypes in our UOT-based Common Class Discovery, we display the mini-batch based similarity matrix and UOT coupling matrix with more detailed mapping data display in Fig.S3 and Fig.S4. To visualize source prototypes, we select the nearest samples to the source prototypes as prototype images. The visualization is conducted on A2W in Office, in which 0-9 classes are common, 10-19 are source-private and 20-30 are target-private. As shown in Fig.S4, the diagonal of UOT coupling matrix of first 10 classes are more clear than others and the others are mapped slightly, which demonstrates that our partial alignment is applicable and accords with the ground-truth labels.

**Qualitative results of target cluster examples.** To better illustrate the target prototypes, we show some examples of top-16 nearest target neighbors of target-common and target-private prototypes. As shown in Fig. S5, we show common mini-clusters of 2 classes per domain. And our OT-based clustering method can help learn a more fine-grained mini-cluster, such as one backpack mini-cluster (left 1) is mainly dark and the other mini-cluster (left 2) is more colorful in Fig. S5(a).

Moreover, we show some examples of target-private clusters in Fig. S6. Although target-private samples lack supervision, our Private Class Discovery can also help them learn representation in a self-supervised manner.