# OpenReview forum: "Unified Optimal Transport Framework for Universal Domain Adaptation"
_NeurIPS.cc/2022/Conference — NeurIPS 2022 Accept_

### Official Review · Reviewer_V2fC · 2022-07-05

**Rating:** 6
**Confidence:** 3
**Soundness:** 2 fair
**Presentation:** 2 fair
**Contribution:** 2 fair

**Summary:**

This paper proposes a novel framework, named UniOT, which jointly considers common class detection and private class discovery in Universal Domain Adaptation (UniDA). UniOT leverages Unbalanced Optimal Transport (UOT) to detect common classes in the target domain without any predefined threshold values. In addition, UniOT learns an OT-based representation learning for private class discovery by considering both global discrimination of clusters and local consistency of samples. Furthermore, the authors introduce a new evaluation metric for the UniDA problem. Finally, experiments show that UniOT outperforms several baselines on UniDA.


**Questions:**

I have the following questions:

1) **Related work.** For the application of OT in (P)DA, partial OT instead of UOT can also be used to allow partial displacement [r1].
[r1] Khai Nguyen, Dang Nguyen, Tung Pham, and Nhat Ho. Improving mini-batch optimal transport via partial transportation. In Proceedings of the 39th International Conference on Machine Learning, 2022.
2) **Figure 2.** The authors should add more description in Figure 2 or its description because it is quite difficult to understand the whole framework at the beginning of the paper. Even after reading the Methodology, it is still time-consuming to figure out different components and their functionalities.
3) **Formulation of OT.** What is the definition of **M** in Equations 2 and 3? Why do the authors convert a minimization problem into a maximization one?
4) **Common class detection.** The authors claimed that this method does not use some manually specified or hand-tuned threshold values to detect common and target-private samples. Though the threshold in Equation (7) is not completely random, users have to tune $\epsilon, \kappa$ for UOT instead. Therefore, it seems like this method only parametrizes the threshold value into other hyper-parameters.
5) **The number of target prototypes.** How is K chosen? Why is it much larger than the actual number of classes?
6) **Equation 13.** Should different notations be used for the prediction of $z_i^t$ and $\tilde{z}_i^t$? Speaking of which, does the prediction of the nearest neighbor of $z_i^t$ lie in the $(B+i)$-th row of $\tilde{Q}^{tt}$ based on the description in L232-233?
7) **Target-private samples classification.**  When is the private class in the target domain classified? L259-260 state that they are all classified as unknown. As far as I understood, UniOT still treats all target-private samples as a single class as in prior works.
8) **Experimental settings.** The authors use a lot of formulas and settings from other papers. It is better to describe them again in the Appendix to improve the readability and make the paper more self-contained.
9) **Baselines' results.** Why are the performance of DANCE in Table 2 and Table 3 worse than weak baselines such as ResNet or DANN? In addition, the authors should add a symbol to indicate which results are taken directly from the original papers and which are the reproduced results.

**Minor:**
* There is a typo in L270. It is either A2D or Amazon to Webcam.
* In column D2A of Table 4, the best H-score is not in the last row so it should not be in bold. In addition, it seems to be unnecessary to additionally report two random tasks for each dataset. It is alright to just report the average performance over all the tasks.

I am happy to increase my score if the above questions are adequately addressed.


**Limitations:**

The authors specified one limitation of their method in Section 5.

**Strengths And Weaknesses:**

In this paper, the authors raise two interesting problems of UniDA that previous works do not consider carefully. Their method to tackle the aforementioned issues is novel to the best of my knowledge. Experimental results back the efficiency of the proposal in UniDA. The authors also conducted adequate ablation studies to verify the importance of different components in their framework. Overall, this paper is well-structured and easy to follow. The main concerns include the clarity of the paper in both methodology and experiment parts.

---

> ### Author Response · Authors · 2022-08-02
> **Authors responses to Reviewer V2fC 1/2**
>
> We thank the reviewer for the positive feedback and constructive suggestions on our paper. We address your detailed comments below:
>
> **Q1**: Related work [R1].
>
> **A1**: Thanks for your suggestion, we have cited this paper and updated the related work in L112-113 in our revised paper.
>
>
> **Q2**: The authors should add more description in Figure 2 or its description to make it more easy to understand.
>
> **A2**: Thanks for your constructive suggestion.
> We have revised the paper to better illustrate our framework.
>
> **Q3**: Formulation of OT. What is the definition of $\mathbf{M}$ in Equations 2 and 3? Why do the authors convert a minimization problem into a maximization one?
>
> **A3**: Following SwAV [7], we adopt maximization objective to formulate OT objective and define $\mathbf{M}$ as a cosine similarity matrix, for which a larger element value indicates a closer distance. But in the original definition of OT in [11], the authors used the distance as cost matrix, for which a smaller element value indicates a closer distance. Therefore, the maximization objective in our paper is equivalent to the minimization one in [11].
>
> **Q4**: Users have to tune $\varepsilon$, $\kappa$ for UOT instead. Therefore, it seems like this method only parametrizes the threshold value into other hyper-parameters.
>
> **A4**: Thank you for raising the concern. Firstly, we would like to justify that the hyper-parameters $\varepsilon$, $\kappa$ in UOT were not tuned but using the default value in JUMBOT solver [13] which can be applied to different applications.
>
> Secondly, we would like to emphasize that existing UniDA methods detect common and target-private samples by using some manually specified or hand-tuned threshold values, which are different for different datasets.  On the contrary, our UniOT can detect common classes without hand-tuned threshold for realistic UniDA. Moreover, the  $\varepsilon$, $\kappa$ in UniOT are identical for all datasets.
>
> **Q5**: Why is it much larger than the actual number of classes? How is K chosen?
>
> **A5**: Thank you for raising the concern.  With regards to the selection of $K$ values, we followed Self-labelling [1] which sets a larger $K$ for larger dataset, for example they set $K=512$ for CIFAR-100.
>
> Secondly, we will explain why $K$ is much larger than the actual number of classes. We would like to point out that our prototype is not the centroid of a class-wise cluster, but a fine-grained mini-cluster, which is has been implemented in cluster-based self-supervised learning [1][7]. Specifically, our OT-based clustering technique in Eq.10 aims at mapping samples to prototypes with equality constraints which enforces samples to be partitioned equally into $K$ mini-clusters, i.e. each mini-cluster has identical size with $\rho$ samples. Then we can roughly determine the number of mini-clusters by $K=n_t/\rho$, where $n_t$ is the number of target samples. Therefore, when $n_t$ is large and $\rho$ is relatively small, $K$ will be large and hence much larger than the actual number of target classes. We hypothesize that each class processes at least about 20 samples, then we can set $K$ by the number of target samples roughly.
>
> **Q6**: Equation 13. Should different notations be used for the prediction of z^t_i and ~z^t_i?
>
> **A6**: Thanks for pointing this typo out.
> We have revised this typo in the revision paper.
>
> **Q7**: **Target-private samples classification.** When is the private class in the target domain classified? L259-260 state that they are all classified as unknown. As far as I understood, UniOT still treats all target-private samples as a single class as in prior works.
>
> **A7**: Thank you for raising the concern.We would like to clarify that we do not classify target-private samples, but we try to exploit and evaluate their cluster representation instead of treating as a single class in both training and testing phase.
>
> In the training phase, we propose Private Class Discovery to discriminate clusters representation among all target samples, including common and target-private samples. In the testing phase, we evaluate NMI of target-private clusters for H$^3$-score, but have to treat all target-private samples unknown to obtain $a_{\bar{\mathcal{C}}_t}$ in H-score and thus can make fair comparison with other methods. To better clarify this, we invite the reviewer to revisit the definition of H-score in Eq.S1 (Appendix) and H$^3$-score in Eq.16.

---

> > ### Author Response · Authors · 2022-08-02
> > **Authors responses to Reviewer V2fC 2/2**
> >
> > **Q8**: Experimental settings. The authors use a lot of formulas and settings from other papers. It is better to describe them again in the Appendix.
> >
> > **A8**: Thanks for your constructive suggestion. We have followed your suggestion and updated Appendix in our revised paper.
> >
> > **Q9**: Baselines' results. Why are the performance of DANCE in Table 2 and Table 3 worse than weak baselines such as ResNet or DANN? In addition, the authors should add a symbol to indicate which results are taken directly from the original papers and which are the reproduced results.
> >
> > **A9**: Thank you for raising the concern.
> > Firstly, we would like to explain that DANCE did not report H-score and thus we reproduce H-score for DANCE for fair comparison. However, H-score is the mainstream evaluation metric for UniDA, which emphasizes the importance of both common and "unknown" accuracy of UniDA methods [18].
> >
> > Secondly, we observe that DANCE got high accuracy $a_{\mathcal{C}}$ for common samples but very extremely low accuracy $a_{\bar{\mathcal{C}}_t}$  for target-private samples (detected as unknown).
> > To explain low accuracy for target-private samples for DANCE, we would like to emphasize that DANCE distinguishes common and target-private classes by a fixed entropy threshold, which is very sensitive to different classes. Furthermore, the entropy metric will fail to discriminate uncertain and extremely sharp predictions, as shown in [16].
> >
> > And thanks for your constructive suggestion of adding a symbol to indicate the source of reported data and we have revised our paper.
> >
> > **Q10**: **Minor.** There are one typo in L270 and one typo in column D2A of Table 4.
> >
> > **A10**: We appreciate the minor suggestions and have revised the paper accordingly.

---

> > > ### Comment · Reviewer_V2fC · 2022-08-06
> > > **Response to Authors**
> > >
> > > I thank the authors for their response.
> > >
> > > After reading their rebuttal, I appreciate that the authors have adequately addressed all of my questions. In addition, the revised version of the paper does improve both the quality and clarity of the paper.
> > >
> > > All in all, I would like to increase my score from 4 to 6.

---

### Official Review · Reviewer_4ds8 · 2022-07-11

**Rating:** 5
**Confidence:** 4
**Soundness:** 2 fair
**Presentation:** 3 good
**Contribution:** 2 fair

**Summary:**

In this work, the authors proposed a unified domain adaptation framework supported by optimal transport in order to transfer among those domains having different categories. Specifically, the authors detect those common classes by predicting the statistics of an unbalanced OT objective. Next, the private class discovery is achieved by doing local alignments in the target domain.

**Questions:**

Please see my comments in the last box. I am willing to improve my score if there are more interpretations and qualitative illustrations.

**Limitations:**

No negative societal impact.

**Strengths And Weaknesses:**

Pros:
- Overall, this paper is well written. The authors provided detailed information about how they designed the training process step by step.
- The authors have provided enough quantitative results in order to demonstrate the effectiveness of their method.

Cons:
 - The proposed method is relatively ad-hoc. The proposed method serves as discriminative losses for supervised learning problems and improves performance in the target domain.
 - Missing discussion on Batch optimal transport. The authors mainly implemented their framework using batch samples. However, It is known that minibatch optimal transport leads to nonoptimal coupling so there should be some discussions on how batch optimal transport affects this method.
 - Qualitative illustrations will be very helpful in illustrating this method's performance. For example, the authors did a great job in creating figure 2 to give a high-level illustration of the framework. It would be great if the authors could give an example of a batch of data, not only the OT couplings but also the corresponding source and target samples/polytopes.

---

> ### Author Response · Authors · 2022-08-02
> **Authors responses to Reviewer 4ds8**
>
> We thank the reviewer for the positive and constructive feedback on our paper.  Here we address your detailed comments below:
>
> **Q1**: The proposed method is relatively ad-hoc. The proposed method serves as discriminative losses for supervised learning problems and improves performance in the target domain.
>
> **A1**: We would like to first clarify that we proposed a unified Optimal Transport (UniOT) framework instead of a unified domain adaptation framework. Notably, the UniOT handles the Common Class Detection and Private Class Discovery in one framework. Therefore, we believe the UniOT framework is not an ad-hoc model.
>
> Secondly, UniOT cannot be simply treated as discriminative losses for supervised learning as it captures the information of common classes between the labelled source and unlabeled target domain and further exploits the inherent data structures in the unlabeled target domain.
>
> Moreover, UniOT can further shed light on other CV or ML tasks. For instance, our proposed Common Class Detection aims at recognizing outlier samples, which can be generalized to open-set recognition [R1], anomaly detection [R2]. Besides, our proposed Private Class Discovery can be also extended to other unsupervised settings for representation learning, such as person-reid [R3], deep clustering[R4].
>
> [R1] Bendale, Abhijit, and Terrance E. Boult. "Towards open set deep networks." Proceedings of the IEEE conference on computer vision and pattern recognition. 2016.
>
> [R2] Chalapathy, Raghavendra, and Sanjay Chawla. "Deep learning for anomaly detection: A survey." arXiv preprint arXiv:1901.03407 (2019).
>
> [R3] Lin, Yutian, et al. "A bottom-up clustering approach to unsupervised person re-identification." Proceedings of the AAAI conference on artificial intelligence. Vol. 33. No. 01. 2019.
>
> [R4] Caron, Mathilde, et al. "Deep clustering for unsupervised learning of visual features." Proceedings of the European conference on computer vision (ECCV). 2018.
>
> **Q2**: Missing discussion on Batch optimal transport. minibatch optimal transport leads to nonoptimal coupling.
>
> **A2**: Thank you for raising the concern. We agree with the reviewer that mini-batch optimal transport leads to non-optimal coupling. Therefore, we followed [7] to utilise the samples from both FIFO memory queue and minibatch to deal with this problem (L199-203, L249-251). We will add more discussion on batch optimal transport and further illustrate our model's advantages over it.
>
> **Q3**: It would be great if the authors could give an example of a batch of data, not only the OT couplings but also the corresponding source and target samples/prototypes?
>
> **A3**: Thanks for positive feedback and constructive suggestion.
> We have followed your suggestion and have updated some qualitative illustrations in our revised supplementary material (Fig.S4, S5, S6 and S7).

---

> ### Author Response · Authors · 2022-08-10
> **Would you please let us know if our qualitative figures are sufficient? Many thanks!**
>
> Dear Reviewer 4ds8,
> Many thanks for your valuable comments and suggestions. As per your suggestions, we have added some qualitative illustrations in our revised supplementary material (Fig.S4, S5 and S6). Would you please let us know if our qualitative figures (Fig.S4, S5 and S6) in the supplementary is sufficient for interpretable illustrations?
>
> By the way, we would like to make a supplement for our previous response on the issues of Batch optimal transport. We agree with the reviewer that mini-batch optimal transport leads to non-optimal coupling and believe the sample number is an important issue for optimal transport. However, we would like to clarify that our UniOT uses not only batch samples but also memory queue (as shown in Fig. 2) to increase the sample number. By this way, more meaningful statistic information can be discovered in the common and private classes. Therefore, our UniOT is not likely to stuck in the nonoptimal coupling compared to the mini-batch optimal transport. The effectiveness of combining the memory queue with batch samples in OT has also been shown in some self-supervised learning methods, such as SwAV [7].
>
> Best regards
>
> The Authors

---

### Official Review · Reviewer_xveW · 2022-07-12

**Rating:** 5
**Confidence:** 5
**Soundness:** 3 good
**Presentation:** 3 good
**Contribution:** 2 fair

**Summary:**

This paper proposes an optimal transport framework to jointly consider common class detection and private class discovery for UniDA. The key point is automatically discovering the intrinsic difference between common and private classes based on the statistical information of the assignment matrix.

**Questions:**

How do you tune the hyperparameters in the method, such as Equation 15? Only sensitivity to one hyper-parameter is stated in the supplementary material.

**Limitations:**

Yes

**Strengths And Weaknesses:**

[Strengths]
This paper addresses Universal Domain Adaptation. The partial alignment for common class detection and representation learning for private class discovery based on UOT and OT models have achieved relatively high classification accuracy. The method of experimental comparison is comprehensive.

[Weaknesses]
The proposition of this method, to automatically discover private specific classes, lacks meaning in practice. Since we do not have the labels of these classes, we cannot use these data, but it brings more complex computing resource consumption. The fundamental goal of UniDA is to improve the precision and recall rate of common classes. Identifying private classes is only a process, so it can be regarded as a class.
The new metric H^3-score is the same problem.

---

> ### Author Response · Authors · 2022-08-02
> **Authors responses to Reviewer xveW**
>
> We would like to thank the reviewer for the constructive comments which helped us improve the quality of our work. In the following, we have provided a point-by-point response to the comments.
>
> **Q1**: The proposition of this method, to automatically discover private specific classes, lacks meaning in practice.  The fundamental goal of UniDA is to improve the precision and recall rate of common classes. Identifying private classes is only a process, so it can be regarded as a class. The new metric H$^3$-score is the same problem.
>
> **A1**:
> Thank you for raising the concern. We completely agree with the reviewer that the fundamental goal of UniDA is to improve the precision and recall rate of common classes. Actually, the motivation of designing Private Class Discovery is to achieve this goal as well.
>
> However, we would like to justify that our design for Private Class Discovery (PCD) is significant for UniDA. Simply treating the private classes as one class will deteriorate the learned representation of the whole target domain and thus may have negative impacts on the common classes. As shown in the following table of averaged recall rate of common classes, our PCD design can bring significant improvement.
>
> |               | Ar2Pr | Cl2Ar | Cl2Pr | Pr2Ar | Rw2Ar | Avg (12 tasks) |
> |---------------|:-----:|:-----:|:-----:|:-----:|:-----:|:--------------:|
> | UniOT w/o PCD | 82.80 | 67.26 | 78.37 | 69.02 | 76.85 |      75.48     |
> | UniOT         | 85.96 | 73.93 | 81.85 | 75.90 | 80.97 |      78.52     |
>
> Moreover, we have added another t-SNE figure in the Fig.S3 (Appendix), which verifies that our PCD does help learn more discriminative representation for common classes and thus improves precision of common classes.
>
> Apart from the performance of common classes, many recent researches [22][16][9] in UniDA also considered the performance of private classes and introduced the H-score metric [16]. Our method with Private Class Discovery can yield high H-score, with 8\% and 7\% improvement on Office and Office-Home respectively (Tab.4). The purpose of PCD is to learn more discriminative feature representation for the whole target domain, including common and target-private samples. This is very beneficial for distinguishing both common and private samples.
>
> We also introduce H$^3$-score, which is a direct and proper metric to evaluate the both the classification of common and representation performance of unknown classes for UniDA. We would like to claim that the goal of domain adaptation is to learn a well-generalized model for target domain with source supervision. In our UniDA case, when target-private dominates the target domain, only learning for extracting common knowledge but ignoring target-private is not a well-generalized model to target domain, since disordered target-private features would likely overlap on common classes representation as we illustrate in Fig.S3 (a) (Appendix).
>
> **Q2**: Since we do not have the labels of these classes, we cannot use these data, but it brings more complex computing resource consumption.
>
> **A2**: Thanks for your comments. However, we respectively do not agree with your statement "Since we do not have the labels of these classes, we cannot use these data". Basically, the whole target domain are unlabeled. Exploiting both the data in labeled source domain and unlabeled target domain is quite standard in the community of UniDA.
>
> With regards to the additional computing resource by Private Class Discovery, the additional computing effort is actually not much. More details about training and inference time consumption please refer to our response to Q1 of Reviewer 1NtB. Apart from the slight increase of computing resource, we would like to emphasize that Private Class Discovery does bring a large improvement: 8\% to Office and 7\% to Office-Home in H-score, compared to training without PCD.
>
>
> **Q2**: How do you tune the hyper-parameters in the method, such as Equation 15? Only sensitivity to one hyper-parameter is stated in the supplementary material.
>
> **A2**: We have conducted the sensitivity analysis for 3 hyper-parameters $\gamma$, $\mu$, $\lambda$ in our Appendix Fig.S1 (b), (c), (d) respectively, which reflects these hyper-parameters are not sensitive. Besides, the coefficient $\varepsilon$ and $\kappa$  are default value from JUMBOT solver [13] which can be applied to different applications.

---

### Official Review · Reviewer_1NtB · 2022-07-13

**Rating:** 7
**Confidence:** 3
**Soundness:** 4 excellent
**Presentation:** 4 excellent
**Contribution:** 4 excellent

**Summary:**

This work is progressing towards solving the Universal Domain Adaptation (UniDA) problem, where we don’t assume any prior knowledge about the target classes. The proposed algorithm, named UniOT, utilizes Optimal Transport (OT) to handle two main issues: i) common class detection and ii) private class discovery.

The common class detection is run by establishing partial alignment between the target features and the source prototypes, while the private class discovery is conducted through aligning the target features with the learnable target prototypes. Both are achieved by solving the OT optimization problem.

From the empirical evaluations on the Office, Office-Home, VisDa, and DomainNet, the proposed algorithm UniOT performs significantly better than the existing UniDA methods indicated by the H-scores. Under varying ratios of common classes, UniOT shows a certain level of robustness compared to DANCE and UAN.

**Questions:**

Still related to the previous point, I'd like to see in more detail about the time complexity of the training and inference.

**Limitations:**

It is implicitly stated in the conclusion that the current approach requires relatively high memory usage and will find a more memory-efficient way.

**Strengths And Weaknesses:**

Strength:
- The OT optimization approach to addressing the UniDA problem by encouraging common class detection and private class discovery is novel and technically sound.
- The performance gain over the existing methods is substantial along the line of UniDA.
- The manuscript is well-written.

Weakness:
- Would be great if information of both training and inference elapsed time is provided, especially at the inference stage as it requires to run an additional OT optimization.

---

> ### Author Response · Authors · 2022-08-02
> **Authors responses to Reviewer 1NtB**
>
> We thank the reviewer for the positive feedback and constructive suggestions on our paper.  Here are our responses to the reviewer's comments.
>
> **Q1**: Elapsed time of the training and inference.
>
> **A1**: We would like to claim that our UniOT can achieve comparable training time (10,000 steps) on different tasks compared to other UniDA methods as the following table shows
> | Method    |   D2A   |  Ar2Cl  |  VisDA  |   S2P   |
> |-----------|:-------:|:-------:|:-------:|:-------:|
> | DANCE[31] | 1h35min | 1h40min | 1h30min | 1h28min |
> | DCC[22]   | 1h38min | 2h16min |    -    |    -    |
> | UniOT     | 1h55min | 1h50min | 1h34min | 1h37min |
>
> Also, the inference time per image is also comparable as the following table shows
> | Method    |   D2A  |  Ar2Cl |  VisDA |   S2P  |
> |-----------|:------:|:------:|:------:|:------:|
> | DANCE[31] | 4.16ms | 3.48ms | 2.23ms | 2.93ms |
> | DCC[22]   | 4.23ms | 3.70ms | 2.26ms | 3.24ms |
> | UniOT     | 4.24ms | 3.57ms | 2.29ms | 3.30ms |
>
> Moreover, to further clarify that an additional OT optimization at the inference stage will not bring heavy burden, we have conducted experiments in a small transfer task in Office and a big one in DomainNet. We observe that it takes 1.84ms in A2D of Office and 8.46ms in P2R of DomainNet. All above experiments were implemented on one NVIDIA TITAN X (Pascal) with 12GB.
>
> **Q2**: Time complexity analysis of OT and UOT optimization.
>
> **A2**: Sinkhorn's algorithm for the OT [11] and UOT [10] has a complexity of order $\widetilde{\mathcal{O}}(n^2)$ [R1][13], where $n$ is the size of cost matrix $\mathbf{M}$ in Eq.2 (assuming $\mathbf{M}$ is a square matrix for the sake of simplicity). In addition, we would like to further point out that many CV and ML tasks [R2][R3] adopt Sinkhorn's algorithm as a lightspeed OT solver.
>
>
> [R1] Pham, Khiem, et al. "On unbalanced optimal transport: An analysis of Sinkhorn algorithm." International Conference on Machine Learning. PMLR, 2020.
>
> [R2] Vocabulary Learning via Optimal Transport for NMT, ACL, 2021.
>
> [R3] Learning to Count via Unbalanced Optimal Transport, AAAI, 2021.

---

### Author Response · Authors · 2022-08-09
**Please let us know if any further questions! Thanks so lot!**

Dear Reviewers and AC,

Thank you for reading our rebuttal. We have tried to address most if not all concerns raised by the reviewers. Please feel free to let us know if you have any further questions.

Thank you and best regards,

Authors

---

### Meta-Review · Area_Chair_CjwK · 2022-08-27

**Recommendation:** Accept
**Confidence:** Certain

**Metareview:**

This work tackles universal domain adaptation, a challenging problem that is usually encountered in real practice. The proposed UniOT algorithm utilizes Optimal Transport to enable common class detection and private class discovery. It is interesting to see that OT can be extended to simultaneously solve these two problems. Reviewers were at the borderline in their preliminary opinions, but after rebuttal and reconsideration, most reviewers acknowledged that their concerns were addressed, and improved their final rating substantially. AC considered the paper itself as well as all reviewing threads, and concluded that the paper has put forward a nice Unified OT framework for the challenging universal domain adaptation problem, yielding promising empirical performance while introducing some nice technical benefits such as the removal of the tedious weight thresholding for outlier class discovery. Thus the paper is recommended for acceptance.

**Award:**

No

---

### Decision · Program_Chairs · 2022-09-14

Accept